



# 1 Modelling bi-directional fluxes of methanol and

# 2 acetaldehyde with the FORCAsT canopy exchange model

**Kirsti Ashworth[1], Serena H. Chung[2], Karena A. McKinney[3*], Ying Liu[3*^], Bill J.**
**Munger[3,4], Scot T. Martin[4] and Allison L. Steiner[1]**
[1] Climate and Space Science and Engineering, University of Michigan, Ann Arbor, MI
48109, USA.
[2] Department of Civil and Environmental Engineering, Washington State University,
Pullman, WA 99164, USA.
[3] Department of Earth and Planetary Sciences, Harvard University, Cambridge, MA 02138,
USA.
[4] School of Engineering and Applied Sciences, Harvard University, Cambridge, MA 02138,
USA.
[*] Formerly of Department of Chemistry, Amherst College, Amherst, MA 01002
[^] Now at Peking University, China
Correspondence to: Kirsti Ashworth (ksashwor@umich.edu)
**Abstract**
The FORCAsT canopy exchange model was used to investigate the underlying mechanisms
governing foliage emissions of methanol and acetaldehyde, two short chain oxygenated
volatile organic compounds ubiquitous in the troposphere and known to have strong biogenic
sources, at a northern mid-latitude forest site. The explicit representation of the vegetation
canopy within the model allowed us to test the hypothesis that stomatal conductance regulates
emissions of these compounds to an extent that its influence is observable at the ecosystem-
scale, a process not currently considered in regional or global scale atmospheric chemistry
models.
We found that FORCAsT could only reproduce the magnitude and diurnal profiles of
methanol and acetaldehyde fluxes measured at the top of the forest canopy at Harvard Forest
if light-dependent emissions were introduced to the model. With the inclusion of such



emissions FORCAsT was able to successfully simulate the observed bi-directional exchange
of methanol and acetaldehyde. Although we found evidence that stomatal conductance
influences methanol fluxes and concentrations at scales beyond the leaf-level, particularly at
dawn and dusk, we were able to adequately capture ecosystem exchange without the addition
of stomatal control to the standard parameterisations of foliage emissions, suggesting that
ecosystem fluxes can be well enough represented by the emissions models currently used.
**Key points:** Canopy exchange model used to probe mechanisms controlling fluxes of
methanol and acetaldehyde; The effects of stomatal control of leaf-level emissions of
methanol and acetaldehyde emissions are not evident at the ecosystem scale; Bi-directional
exchange of oxygenated volatile organic compounds can be simulated by models that
explicitly and holistically consider canopy processes

## 1  Introduction

The exchange of many oxygenated volatile organic compounds (oVOCs) from forest canopies
has recently been observed to be bi-directional, with periods of strongly positive (i.e. up out
of the canopy to the atmosphere above) and negative (i.e. downward) fluxes (Park et al.,
2013; Karl et al., 2005; McKinney et al., 2011). Several of these compounds, e.g. acetone,
acetaldehyde, and methanol, are present in the atmosphere in large quantities (Singh et al.,
1995; Heikes et al., 2002; Millet et al., 2010; Jacob et al., 2002). They are also chemically
active, with acetone and acetaldehyde leading to the formation of PAN (peroxyacetyl nitrate)
and the transport of reactive nitrogen to remote regions (Fischer et al., 2014), and methanol
contributing significantly to the production of ground-level ozone (Tie  et al., 2003). These
oVOCs have potentially important implications for regional air quality and climate modelling
and for estimating global atmospheric burdens of many trace gases (e.g. Folberth et al., 2006;
Fischer et al., 2014). However, many regional and global atmospheric chemistry and transport
models (CTMs) do not explicitly include dynamic biogenic sources and sinks of oVOCs.
While most now incorporate on-line calculations of biogenic emissions of isoprene and
monoterpenes, based on the light and temperature-dependence algorithms developed by
Guenther et al. (1995; 2006; 2012), methanol emissions have only been recently included in
some CTMs (e.g. GEOS-Chem; Millet et al., 2010; Laboratoire de Météorologie Dynamique
zoom (LMDz): Folberth et al., 2006) and most still rely on non-dynamic emissions
inventories for methanol and acetaldehyde if primary biogenic emissions of these species are
included (e.g. UKCA: O'Connor et al., 2014). Furthermore, Ganzeveld et al. (2008)



demonstrated the weaknesses of the algorithms currently used in 3-D chemistry transport
models to calculate primary emissions of methanol on-line. Similarly, dry deposition schemes
in CTMs are usually based on fixed deposition velocities (Wohlfahrt et al., 2015) or
calculated from roughness lengths and leaf area index values assigned to generic landcover
types (e.g. FRSGC-UCI: Wild et al., 2007; LMDz: Folberth et al., 2006). This simplistic
approach to biogenic sources and sinks may be a critical omission limiting their capability of
accurately simulating atmospheric composition in many world regions.
Here we focus on methanol and acetaldehyde, two oVOCs that are frequently observed in and
above forests but whose sources, sinks and net budgets are not known with any certainty
(Seco et al., 2007; Niinemets et al., 2004). While biogenic sources of both are strongly
seasonal, fluxes and concentrations can remain high throughout the growing season
(Stavrakou et al., 2011; Millet et al., 2011; Karl et al., 2003; Wohlfahrt et al., 2015).
Methanol fluxes are on the same order of magnitude as isoprene at many sites in the US (Fall
and Benson, 1996), suggesting their regional and global importance. The fundamental
mechanisms leading to the synthesis and/or subsequent release of methanol and acetaldehyde
are not currently fully understood (Karl et al., 2002; Seco et al., 2007).
Methanol is known to be produced from demethylation processes during cell wall expansion
and leaf growth with emissions peaking during springtime leaf growth and declining with leaf
age (Fall and Benson, 1996). The factors controlling its subsequent release to the atmosphere
are harder to decipher (Huve et al., 2007; Niinemets et al., 2004). Measurements at all scales
from leaf-level to branch enclosure and ground-based ecosystem-scale field measurements
(e.g. Kesselmeier et al., 2001; Karl et al., 2003; Seco et al., 2015; Wohlfahrt et al., 2015), as
well as satellite inversions (e.g. Stravakou et al., 2012) demonstrate a strong diurnal profile of
methanol fluxes similar to that of isoprene (e.g. Fall and Benson, 1996). Methanol synthesis,
unlike that of isoprene, is not specifically linked to photosynthesis and the light-dependence
observed in leaf-level emissions have been shown to result from regulation by the stomata due
to the high solubility of methanol in water (e.g. Nemecek-Marshall et al., 1995; Niinemets
and Reichstein, 2003a,b; Huve et al., 2007).
The pathways leading to both the synthesis and emission of acetaldehyde are not clear (Karl
et al., 2002; Jardine et al., 2008). Acetaldehyde has long been known to be an oxidation
product of ethanol produced in leaves under anoxic conditions (Kreuzwieser et al., 2000) but
this cannot explain the strong emissions observed under normal environmental conditions at



mid-latitude forests (e.g. Seco et al, 2007; Karl et al., 2003). Karl et al. (2003) observed that
bursts of acetaldehyde were emitted during light-dark transitions and postulated that such
emissions were associated with pyruvate decarboxylation. Leaf-level measurements of
acetaldehyde emissions have also been found to be tightly coupled to stomatal aperture (e.g.
Kreuzwieser et al., 2000; Karl et al., 2002; Niinemets et al., 2004) and it has been suggested
that this may account for observed light-dependent ecosystem-scale emissions of
acetaldehyde (Jardine et al., 2008).
Previous studies have suggested that the role of stomatal conductance in determining net flux
of oVOCs could be incorporated in large-scale models by adopting a compensation point
approach (see e.g. Harley et al., 2007; Ganzeveld et al., 2008; Jardine et al., 2008). The
compensation point for a given compound is the atmospheric concentration of that compound
at which the leaf, plant or canopy switches from acting as a net source to a net sink. While
firmly based in plant physiology and plant response to environmental conditions, this
approach would allow models lacking leaf-level processes to account for the changes in flux
direction (Harley et al., 2007; Ganzeveld et al., 2008). Observational (Jardine et al., 2008) and
modelling studies (Ganzeveld et al., 2008) have both shown the potential power of this
approach, although Jardine et al. (2008) found that the compensation point was heavily
dependent on light and temperature and may therefore not be straightforward to implement.
Here we use the FORCAsT (FORest Canopy-Atmosphere Transfer) canopy-atmosphere
exchange model (Ashworth et al., 2015) to investigate the key processes driving fluxes of
methanol and acetaldehyde, and explore possible underlying causes of their bi-directional
exchange. The model represents all within-canopy processes: primary emissions, chemical
and photolysis reactions, turbulent mixing and deposition. A particular strength of the
FORCAsT model is the inclusion of plant processes relevant to photosynthesis and
respiration; stomatal conductance is explicitly calculated by FORCAsT. We therefore focus
on exploring the role of primary biogenic emissions of methanol and acetaldehyde on canopy-
top fluxes. We assess the effectiveness of different representations of bVOC emissions
mechanisms in capturing ecosystem-scale fluxes. For the first time in a canopy exchange
model, we implement a mechanism by which stomatal conductance explicitly regulates
primary emissions in order to assess its role in governing primary emissions and influencing
ecosystem-scale bi-directional exchange of these key oVOCs. We compare modelled fluxes
using this mechanism with those from traditional empirical algorithms for direct and storage



emissions and with fluxes measured just above the top of the canopy at Harvard Forest in July

2   2012.

## 2   Methods

### 2.1   Harvard Forest measurements

Harvard Forest is situated in a rural area of Massachusetts, approximately 90 km from Boston
and 130 km from Albany. It is classified as a mixed deciduous broadleaved forest, with red
oak (36%) and red maple (22%) as the dominant species (McKinney et al., 2011). Continuous
measurements of micro-meteorological variables and air pollutants have been made from the
Environmental Monitoring Station (EMS) Tower, part of the AmeriFlux network, for 25 years
(Munger and Wofsy, 1999a;b). The tower, located at 42.5°N and 72.2°W and an elevation of
340 m, is 30 m high and is surrounded by primary forest with an average height of around 23
m. The long-term meteorological measurements include photosynthetically active radiation
(PAR), relative humidity (RH) and air temperature at multiple heights on the tower, together
with wind speed and direction recorded just below the top of the tower (at ~29 m) (Munger
and Wofsy, 1999a). In addition to exchanges of $CO_2$ collected to assess photosynthetic
activity and productivity, concentrations of CO at the top of the tower and fluxes of $O_3$ (at
multiple heights on the tower) are also routinely measured (Munger and Wofsy, 1999c). NO
and $NO_2$ concentrations and fluxes have been recorded in the past, with the most recent
measurements in 2002. In addition to these continuous atmospheric measurements, a suite of
other data is gathered periodically to determine ecosystem health and functioning. Such data
include leaf area index, tree girth, litter mass, leaf chemistry, and soil moisture and respiration
(Munger and Wofsy, 1999b).
Concentrations and fluxes of bVOCs and their oxidation products have also been measured at
the EMS Tower during several summer growing seasons (McKinney et al, 2011; Goldstein et
al., 1999; 1995), augmenting the Ameriflux suite of observations. Between 7th June and 24th
September 2012, a proton-transfer-reaction time-of-flight mass spectrometer (PTR-TOF-MS
8000, Ionicon Analytik GmbH, Austria) was used to measure the concentrations of volatile
organic compounds at the site. The PTR-TOF-MS is capable of the rapid detection of
hundreds of different VOCs at concentrations as low as a few pptv. PTR-TOF-MS has been
described previously by Jordan et al. (2009a, b) and Graus et al. (2010). The instrument
utilizes a high-resolution TOF detector (Tofwerk AG, Switzerland) to analyze the reagent and




product ions and allows for exact identification of the ion molecular formula (mass resolution
>4000).
Ambient air was sampled from an inlet mounted at the top of the 30-m EMS tower at a total
flow rate of 5 slpm using a configuration identical to that used by McKinney et al. (2011) in
2007. $H_3O^+$ reagent ions were used to selectively ionize organic molecules in the sample air.
The instrument was operated with a drift tube temperature of 60°C and a drift tube pressure of
2.20 mbar. The drift tube voltage was set to 550 V, resulting in an E/N of 126 Td (E, electric
field strength; N, number density of air in the drift tube; unit, Townsend, Td; 1 Td = $10^{-17}$ V
$cm^2$). PTR-TOF-MS spectra were collected at a time resolution of 5 Hz. Mass calibration was
performed every 2 min with data acquisition using the Tof-Daq v1.91 software (Tofwerk AG,
Switzerland). A calibration system in which gas standards (Scott Specialty Gases) were added
into a humidified zero air flow at controlled flow rates was used to establish the instrument
sensitivities to VOCs. Every 3 h the inlet flow was switched to pass through a catalytic
converter (platinum on glass wool heated to 350°C) to remove VOCs and establish
background intensities.
The PTR-TOF-MS captures the entire mass spectrum in each 5-Hz measurement, providing a
continuous mixing ratio time series at each mass-to-charge ratio rather than the disjunct time
series obtained in previous PTR-MS studies at this site (McKinney et al., 2011). As a result,
direct, rather than virtual disjunct, eddy covariances were determined and are reported herein
(Mueller et al., 2010). Wind speeds recorded at 8 Hz by a tri-dimensional sonic anemometer
located at the same height and less than 1 m away from the gas inlet were averaged to a 5-Hz
time base, synchronized with the mixing ratio data, and used in the eddy covariance
calculations. Eddy covariance fluxes were calculated from the data for 30-minute intervals
using methods described in McKinney et al. (2011). Ambient mixing ratios were averaged
over the same 30-minute intervals for which fluxes were calculated. The 30-minute average
mixing ratios and fluxes were then binned by time of day to calculate diurnal averages.
Eddy covariance is a powerful technique for the direct detection and estimation of ecosystem-
scale fluxes of trace gases within and above vegetation canopies (see reviews by Baldocchi,
2003; 2014). However, its reliability for measuring night-time fluxes can be low (Gu et al.,
2005; Baldocchi, 2014; Goulden et al., 1996; Jarvis et al., 1997). Its successful application
relies on assumptions of steady-state conditions, conditions that do not always exist at night
(see e.g. Baldocchi, 2003). The night-time formation of a stable atmospheric layer near the





surface can result in stratification, trapping trace gases below the instrument detection height
and altering the footprint of the flux measurement (Gu et al. 2005; Baldocchi, 2003) leading
to high associated errors in flux estimation (Goulden et al., 1996). While we acknowledge that
the magnitudes of the recorded night-time fluxes during summer 2012 may have large
associated errors, we are confident in the direction of the exchange as we see variation
between different species suggesting no systematic bias.
Isoprene, total combined monoterpenes, MVK and MACR (detected as a single combined
species), methanol, acetaldehyde and acetone were all detected at concentrations well above
the PTR-MS detection limit and determined to be free from interference from other
compounds (McKinney et al., 2011). Here we confine our analysis to concentrations and
fluxes of methanol and acetaldehyde. Table 1 summarises the relevant flux, concentration and
meteorological measurements made at the EMS tower during the summer of 2012.

## 2.2  FORCAsT1.0 canopy exchange model

FORCAsT (version 1.0) is a single column (1-D) model that simulates the exchange of trace
gases and aerosols between the forest canopy and atmosphere. A full description of
FORCAsT is given in Ashworth et al. (2015). Here we provide a brief overview, summarise
biogenic emissions and flux calculations in the model and describe the simulations performed.
FORCAsT1.0 has 40 vertical levels of varying thickness extending to a height of ~4 km, with
the highest resolution nearest the ground where the complexity is greatest, i.e. within the
canopy space. Micro-meteorological conditions (temperature, PAR, RH) within the canopy
are determined prognostically by energy balance, accounting for the physical structure of the
canopy. The gas-phase chemistry scheme incorporated in FORCAsT1.0 is a modified version
of the CalTech Chemical Mechanism (CACM; Griffin et al., 2002; 2005; Chen and Griffin,
2005), which includes 300 species whose concentrations are solved at every chemistry
timestep (currently 1 minute), plus $O_2$ and water vapour (Ashworth et al., 2015). Ninety-nine
of the species are assumed to be condensable, and are lumped into 11 surrogate groups based
on similar volatility and structure. Aerosol-phase concentrations of these surrogate groups are
also calculated at every timestep based on equilibrium partitioning (Ashworth et al., 2015;
Chen and Griffin, 2005).
The CACM chemistry mechanism in FORCAsT treats methanol as an individual species,
although its reactions are limited to oxidation by OH to produce formaldehyde. Acetaldehyde





is not treated individually but is instead mapped to a lumped group of aldehydes (ALD1, with
$<C_5$). The oxidation reactions for this group are based on acetaldehyde and no other species is
currently emitted into the ALD1 group. Acetaldehyde has a far greater number of chemical
sources and sinks in the FORCAsT simulations of a forest environment than methanol. See
Ashworth et al. (2015) for details of the reactions and reaction rates included in FORCAsT.
FORCAsT incorporates dry deposition of all species based on the resistance scheme of
Wesely (1989) and modified by Gao et al. (1993). The scheme assumes that the rate of
deposition of a compound to canopy surfaces is determined by atmospheric, boundary and
surface resistances operating in series or parallel analogous to electrical resistances.
Atmospheric and surface boundary layer resistances are common to all chemical species and
are dependent on turbulence. As FORCAsT includes an explicit representation of the canopy,
the surface resistance term includes cuticular, mesophyllic and stomatal resistances which are
dependent on the physic-chemical properties of the depositing species as well as the light,
temperature and water potential of the leaf. The deposition scheme described in Ashworth et
al. (2015) and Bryan et al. (2012) has been updated to include methanol. The deposition
velocity of acetaldehyde is calculated using parameters for the lumped ALD1 group, and the
parameters for ALD1 and methanol deposition are shown in Table 3.
While a 1-D model cannot capture horizontal transport, FORCAsT does include a simple
parameterisation to account for advection (Bryan et al., 2012; Ashworth et al., 2015). For the
simulations here, only advection of $NO_2$ is considered such that a $NO_2$ mixing ratio of 1 ppbv
is set just above the canopy based on average midday (defined as 10:00-17:00 EST) $NO_x$ and
$NO_y$ (total reactive nitrogen species) concentrations. While nitrogen species were not
measured at Harvard Forest in 2012, concentrations reported from the site by Moody et al.
(1998) are extrapolated to 2012 using July monthly average $NO_x$ levels measured at the
nearby US EPA monitoring station at Ware 42.3ºN, 72.3ºW, elevation 312 m (roughly 30 km
southwest of the EMS Tower). This scaling accounts for the observed decrease in $NO_x$ levels
across the region as a result of emission reduction strategies (see e.g. EPA, 2015). All $NO_x$ is
assumed to be advected as $NO_2$. The initial concentration of $N_2O_5$ at 29 m was set to give an
average $NO_x$:$NO_y$ ratio of 0.4 (Moody et al., 1998), assuming all residual $NO_y$ to be $N_2O_5$
initially.





### 2.2.1 Flux calculations

Fluxes of gases and particles are calculated to be proportional to both the concentration gradient and the efficiency of vertical mixing between adjacent model layers (Eq. 1). Upward fluxes are modelled as positive and occur when the concentration of a particular species is higher at a lower height. The flux, $F_i$ (kg m$^{-2}$ s$^{-1}$) of an individual species, $i$, between two model levels is given by:

$$F_i = -K_H \frac{\Delta C_i}{\Delta z}, \qquad (1)$$

where $K_H$ is the eddy diffusivity (m$^2$ s$^{-1}$), $\Delta C_i$ the difference in mass concentrations (kg kg$^{-1}$) at the mid-height of the levels, and $\Delta z$ the difference in height (m) between the levels. Eddy diffusivity, concentrations of all gas-phase and aerosol species, and fluxes are calculated at 1-minute timesteps. The eddy diffusivity at the instrument height of 29 m is constrained by observed windspeeds (Bryan et al., 2012).

Modelled fluxes should be viewed as an instantaneous snapshot, both temporally and spatially, as the calculation relies heavily on the concentration gradient across an arbitrary boundary level, in this case the instrument height of 29 m. Actual concentration gradients display rapid fluctuations (see e.g. Steiner et al., 2011) due to heterogeneity in emissions (see e.g. Bryan et al., 2015) and chemistry (see e.g. Butler et al., 2008), as well as the occurrence of coherent structures which can result in counter-gradient flow of matter (Steiner et al., 2011 and references therein).

### 2.2.2 Biogenic emissions

Emissions of VOCs from vegetation can be described as following one of two possible routes (Grote and Niinemets, 2008). In the first, the compound is released to the atmosphere immediately on production (e.g. isoprene). Such emissions are tightly coupled to photosynthesis and are therefore dependent on both temperature and light, falling to zero at night. We refer to such emissions as "direct". In the second pathway, VOCs are stored in specialist structures within the plant after their production (e.g. monoterpenes). Emissions from these storage pools occur by diffusion and are controlled by temperature alone. We term these "storage" emissions. It is thought that emissions of oVOCs are a combination of these ("combo"), with a proportion released directly on synthesis and the remaining fraction emitted from storage pools.



Emission rates are calculated in FORCAsT by modifying basal emission factors (rates at
standard conditions, usually 30°C and 1000 µmol $m^{-2}$ $s^{-1}$ of PAR) according to empirical
relationships describing their dependence on light and temperature. These modifications
(referred to as activity factors) follow the standard parameterisations of Guenther et al. (1995;
2012). For storage emissions, which are modelled as dependent on temperature only, the
activity factor is a simple exponential relationship:
$\gamma_T = e^{-\beta(T_L - T_S)}$,    (2)
where $\gamma_T$ is the temperature-dependent activity factor for storage emissions, $\beta$ the temperature
response factor ($K^{-1}$), $T_s$ is 293K, $T_L$ (K) the leaf temperature (see Guenther et al., 2012). For
further details of the activity factors for direct emissions included in FORCAsT the reader is
referred to Ashworth et al. (2015) and references therein.
### 2.2.3 Stomatal resistance
FORCAsT includes a physical representation of a forest canopy, with the lowest eight model
levels set as trunk space and the next ten as crown space. The ten crown space levels contain
the foliage; the total leaf area estimated for 2012 based on litter fall is distributed among the
levels according to balloon measurements made at the site by Parker (1999). Within each
crown space level, the leaves are assigned to one of nine equally-spaced angle classes
assuming a spherical canopy based on leaf normal angle (Goel et al., 1989) and the fraction of
shaded leaf area calculated. Photosynthetic parameters, including stomatal resistance, are then
calculated for each leaf angle class at each level within the crown space. The stomatal
conductance (inverse of stomatal resistance) describes the aperture of the stomata and
determines evapo-transpiration (hence heat flux and energy balance) and deposition rates
within FORCAsT. It is not currently used to control the rate of biogenic emissions.
Stomatal resistance is modelled according to leaf temperature, PAR, water potential and
vapour pressure deficit using the relationships developed by Jarvis (1976) as described by
Baldocchi et al. (1987). The overall stomatal resistance ($r_s$) is the product of these individual
factors (Eq. 3) which are summarised below in Eqs. 4-8
$R_s = r_{smin} \cdot r_s(PAR) \cdot r_s(T) \cdot r_s(D) \cdot r_s(p)$,    (3)
where $r_s(PAR)$ is the response of stomatal resistance to changes in PAR, $r_{smin}$ (s $m^{-1}$) is the
minimum stomatal resistance and $b_{rs}$ is an empirical coefficient:





$r_s(\text{PAR}) = r_{\text{smin}}\left(1 + \frac{b_{\text{rs}}}{\text{PAR}}\right),$ (4)
and $r_s(T)$ is the response of stomatal resistance to changes in leaf temperature ($T_{\text{lf}}$; °C), $T_{\text{min}}$,
$T_{\text{max}}$, and $T_0$ are the minimum and maximum temperatures for stomatal opening and optimum
temperature respectively:
$r_s(T) = \left\{\left(\frac{T_{\text{lf}} - T_{\text{min}}}{T_0 - T_{\text{min}}}\right)\left(\frac{T_{\text{max}} - T_{\text{lf}}}{T_{\text{max}} - T_0}\right)^{b_T}\right\}^{-1},$ (5)
$b_T = \left(\frac{T_{\text{max}} - T_0}{T_{\text{max}} - T_{\text{min}}}\right),$ (6)
and $r_s(d)$ is the relationship between stomatal resistance and vapour pressure deficit ($D$;
mbar), and $b_v$ is an empirical coefficient:
$r_s(D) = \left(1 + \frac{b_v}{D}\right)^{-1},$ (7)
Water potential is assumed to act only once a threshold value is reached. Above this value it is
modelled as:
$r_s(\varphi) = \left(\frac{1}{a \cdot \varphi + b_w}\right),$ (8)
where $\varphi$ is the water potential (bar), and $a$ and $b_w$ are constants. Below the water potential
threshold $r_s(\varphi)$ is taken as unity. The values of the constants used in these calculations are
shown in Table 4.
Stomatal resistance is only calculated in FORCAsT during the day (defined within FORCAsT
as PAR $\geq 0.01$ W m$^{-2}$); at night stomatal resistance is assumed equal to the minimum cuticular
resistance (3000 s m$^{-1}$).
**2.3   FORCAsT simulations**
All model simulations were performed for an average day in July 2012, the middle of the
growing season, to ensure measurement data did not include either the spring burst of
methanol nor elevated acetaldehyde emissions during senescence. FORCAsT was initiated
with site-specific parameters and measurements of the physical structure of the canopy and
environmental conditions (Table 2). Initial meteorological conditions and atmospheric
concentrations of chemical species were taken from the 2012 EMS tower data (see Table 2).
Initial air temperature above the canopy is calculated on-line using the average lapse rate
observed by the radiosonde at Albany (the nearest sounding station, ~90 km from Harvard



Forest), and within the canopy by interpolation with the 2-m temperature reading.
Concentrations of $O_3$ within the canopy are based on observations from the EMS tower, and
above the canopy follow a typical night-time profile as described in Forkel et al. (2006).
Concentrations of other species are assumed to decay exponentially with height such that the
e-folding height is 100 m for short-lived species and 1000 m for longer-lived compounds. All
model simulations started at 00:00 EST and continued for 48 hours, with the same driving
data used for each 24 hour period and analysis confined to the second day to account for
model spinup.
In addition to a baseline simulation, we perform a series of simulations that represent the
potential bVOC emissions routes using the "traditional" algorithms based on the observed
light and/or temperature dependence encapsulated in the MEGANv2.1 model of Guenther et
al. (2012); see Section 2.2.2. We then introduce stomatal control to the temperature-only
dependent emissions (i.e. those from storage pools) to determine whether the observed leaf-
level regulation of the emissions of oVOCs by stomatal aperture affects ecosystem-scale
fluxes (Section 2.3.3). A final series of sensitivity tests explores the extent to which stomatal
control governs canopy-top fluxes (Section 2.3.3). Table 5 summarises the simulations and
sensitivity tests.
Model performance was evaluated against average fluxes and concentrations measured at 29
m throughout July 2012 at Harvard Forest. The raw measurement data were grouped and
averaged for each model output time for the duration of the campaign period to create
"typical" diurnal profiles of methanol and acetaldehyde fluxes and concentrations. The flux
data in particular exhibited large variability introducing high uncertainty to the assessment.
Observations of both fluxes and concentrations of acetaldehyde were more variable than those
of methanol, reflecting the greater number of chemical sources and sinks of acetaldehyde in
conjunction with lower emission rates. The observations referred to throughout the main text
and shown in Fig. 4, 6 and 7 are these averages of the campaign data.

### 2.3.1 Baseline

All simulations were driven using meteorology for an average July day with initial conditions
set to July average values for all variables at 00:00 EST (shown in Table 2). For the baseline
simulation, default FORCAsT settings for emissions, dry deposition and chemical production
and loss (Ashworth et al., 2015) were used; the default FORCAsT settings do not consider



primary emissions of methanol and acetaldehyde. Only primary emissions of isoprene and the
monoterpenes α-pinene, β-pinene and *d*-limonene are included in the base case, with emission
factors (Table 6a) based on average rates for mixed deciduous woodland in N America
(Geron et al., 2000; Helmig et al., 1999).

**2.3.2  Primary emissions sensitivity tests**

Simulations including primary emissions of methanol and acetaldehyde were conducted to
understand the effect of adding primary emissions of oVOC. The specific changes from the
baseline are described below and summarised in Table 5.
In the first three "emissions" (E-) simulations, primary emissions of methanol and
acetaldehyde are included: firstly with all emissions assumed to be direct (E-direct), then all
from storage pools (E-storage), and finally as a combination of the two with 80% taken to be
direct and the remainder storage (E-combo). Emission rates for methanol and acetaldehyde
(Table 5) were initially based on standard emission factors for methanol and bidirectional
VOCs, respectively, for temperate deciduous broad-leaved trees given by Guenther et al.
(2012) and scaled for this site by isoprene emission factor. The emission factors were then
modified to best reconcile modelled and observed concentrations and fluxes at 29 m whilst
ensuring that total canopy emissions for all simulations were within ±10%. The proportion of
80% direct and 20% storage emissions included in E-combo was also based on the "light-
dependent fractions" assigned to methanol and bidirectional VOCs by Guenther et al. (2012).
A sensitivity test with the combination of 90% direct and 10% storage (E-combo90) was also
performed. For each simulation, emission factors and total emissions are listed in Table 6b,
and diel profiles of total emissions, deposition and canopy chemical production and loss are
shown in Fig. 1. While the general pattern of emissions is the same in all simulations (Figs.
1a,b), the magnitude of the midday peak and overnight emission rate vary between the
different emission pathways introduced. The greater the contribution from storage the higher
the overnight and the lower the daytime peak with E-direct (green line; 0% storage emissions)
and E-storage (blue line; 100% storage) representing the extreme cases. Changes in emission
rates alter the concentrations of methanol or acetaldehyde within the crown space driving
differences in both dry deposition (Figs. 1c,d) and chemical production and loss (Figs. 1e,f)
rates. Fig. 1 further demonstrates the relatively small contribution of chemical production and
loss to the canopy space budgets of methanol and acetaldehyde.





### 1  2.3.3  Stomatal control sensitivity tests

Previous theoretical and laboratory-based studies have demonstrated the importance of
stomatal aperture in the regulation of emissions of oVOCs from storage structures (e.g.
Niinemets and Reichstein, 2003a,b; Nemecek-Marshall et al., 1995; Huve et al., 2007; Karl et
al., 2002). Controlled experiments and leaf-level measurements suggest that emissions of
many VOCs are dependent on stomatal conductance, although the extent to which the stomata
regulate emission rates is highly dependent on both the compound and the leaf structure
(Niinemets and Reichstein, 2003a).
Further sensitivity tests were performed specifically to test the dependence of the emissions of
methanol and acetaldehyde on stomatal conductance. Stomatal resistance (the reciprocal of
conductance) is explicitly calculated for every canopy level at every model timestep based on
incident PAR, leaf temperature and water potential (Eq. 3). In this series of tests, the
calculated resistances were used to scale the temperature-dependence of storage emissions of
methanol and acetaldehyde (given in Eq. 2) for both the storage and combo emission
pathways as shown in Eq. 9.
$$\gamma_{\mathrm{TR}} = \gamma_{\mathrm{T}} \cdot R_{\mathrm{fct}} = e^{-\beta(T_{\mathrm{L}} - T_{\mathrm{S}})} \cdot R_{\mathrm{fct}} \, , \tag{9}$$
where $R_{\mathrm{fct}}$ is a stomatal control factor.
In the first of the "stomatal control" (S-) sensitivity tests, $R_{\mathrm{fct}}$ increased proportionally with
stomatal conductance (i.e. inversely with stomatal resistance) as shown in Eq. 10:
$$R_{\mathrm{fct}} = \frac{3000}{n \cdot R_{\mathrm{stom}}}, \tag{10}$$
where $R_{\mathrm{stom}}$ ((μmol m$^{-2}$ s$^{-1}$)$^{-1}$) is the stomatal resistance, 3000 is the limiting night-time value
of $R_{\mathrm{stom}}$ and $n$ is a scaling factor, which was initially set to 3 for the S-storage and S-combo
simulations. The effect of the choice of value of $n$ is explored in Section 3.5.
Fig. 2 shows the diel cycle of stomatal resistances calculated in FORCAsT for each model
level within the crown space; an average canopy resistance is also indicated. $R_{\mathrm{stom}}$ is set to
3000 overnight and falls to a minimum during the middle of the day when light levels are
highest in the canopy. $R_{\mathrm{stom}}$ is lower at the top of the canopy and increases with increasing
depth into the foliage layers. The profile of $R_{\mathrm{fct}}$ (Eq. 10) describes the inverse of $R_{\mathrm{stom}}$,
reaching a peak at midday and having a greater value higher in the canopy. As shown in the
middle panels $R_{\mathrm{fct}}$ reaches >1.0 during the middle of the day for all but the very lowest canopy





layers. Modelled stomatal control (S- simulations) therefore enhances emissions of methanol
and acetaldehyde above those simulated by traditional emissions algorithms during this time.
There is evidence that this may be biologically realistic with stomatal aperture limiting
emissions from storage pools and leading to increased pool size and hence greater
concentration gradients between plant tissue and the surrounding atmosphere (see e.g. Jardine
et al., 2008). This in turn drives an increase in emissions above those predicted based on
synthesis rates of oVOC. However, traditional emissions models were derived to fit observed
emission rates (see e.g. Guenther et al., 1993) and could be assumed to account for this effect.
Hence, a second set of "modified" stomatal control (R-) experiments was performed in which
it was assumed that beyond a threshold stomatal aperture, stomatal conductance no longer
controls emissions, which continue unhindered once the stomates are considered to be fully
open.  Beyond this threshold emissions from storage pools are regulated by temperature alone
according to the relationship in Eq. 2, i.e. $R_{fct}$ in Eq. 9 takes a value of unity, thus assuming
that "traditional" emissions algorithms correctly capture emission rates during the middle of
the day. Within FORCAsT this was modelled using a threshold function:
$$R_{\text{fct}} = \frac{3000}{n \cdot R_{\text{stom}}}, R_{\text{fct}} < 1.0 \tag{11a}$$
$$R_{\text{fct}} = 1.0, \quad \text{at all other times} \tag{11b}$$
The use of the function shown in Eqs. 11a and 11b limits the temporal extent of stomatal
control on methanol and acetaldehyde emissions for most canopy layers to the transition times
of day (dawn and dusk) when the stomata are either opening or closing as light levels increase
or decrease. This is consistent with results from controlled experiments and observations by
Niinemets and Reichstein (2003a) that indicate that stomatal aperture has only a transient
effect on the emissions of oVOC and is negligible under steady-state light conditions. It
should be noted however that under the average July radiation conditions the lower canopy
levels do not receive sufficient PAR to reach this threshold value within FORCAsT.
**3    Results**
**3.1    Summary of observations**
July was roughly the middle of the growing season in 2012 with emissions unaffected by
springtime leaf flush or autumn senescence. As observed previously at many sites, fluxes of
both methanol and acetaldehyde are highly variable with periods of net positive and net



negative exchange (e.g. McKinney et al., 2011; Wohlfahrt et al., 2015; Karl et al., 2005). In
prior years, concentrations of methanol at Harvard Forest remained high even outside of the
spring emissions peak (McKinney et al., 2011).
Fig. 3 shows correlations of the observed daytime (05:00-19:00 EST) fluxes of methanol and
acetaldehyde during July 2012 with air temperature, PAR, canopy stomatal conductance, and
concentrations of methanol and acetaldehyde respectively. Canopy stomatal conductance for
the tower footprint was estimated from energy fluxes measured at Harvard Forest following
the methodology of Shuttleworth et al. (1984) to calculate surface resistances. The raw data
were highly scattered, and were therefore binned by the independent variable in each case
with Fig. 3 showing only the mean values (with bars showing ±1 standard deviation to give an
indication of the variability of the data) for each of these bins for clarity. The weak
relationships with each of the environmental variables evident in Fig. 3 illustrate the difficulty
in identifying the key processes driving canopy-scale exchanges of oVOC under varying
environmental conditions from observations alone.
Canopy-top fluxes of methanol appear to be positively correlated with temperature (Fig. 3a)
and to a lesser extent with PAR (Fig. 3c). The correlation with temperature seems to be
exponential as might be expected. The contribution of stomatal conductance to observed
methanol fluxes is more difficult to interpret although the data appear to show a strong linear
correlation at low conductance, suggesting that at small stomatal aperture the stomata exert
control over fluxes of methanol to the extent that it is observable at the canopy scale. A
similar relationship between canopy-top methanol fluxes and concentrations is likely due to
the influence of atmospheric concentrations on dry deposition to surfaces within the canopy.
Fluxes of acetaldehyde are lower and more variable than those of methanol, and averages are
clustered near zero. However, they do appear to be positively correlated with temperature
(Fig. 3b) although the relationship is weaker and does not appear to be exponential. There is
no discernible correlation between acetaldehyde fluxes and either PAR (Fig. 3d) or stomatal
conductance (Fig. 3f). This might suggest that acetaldehyde emissions are not controlled by
stomatal aperture but may rather indicate the influence of the greater number of sources and
sinks for acetaldehyde at the spatial and temporal scale of the canopy. Jardine et al. (2008)
describe a clear negative correlation between acetaldehyde fluxes and concentrations
measured in the laboratory and Fig. 3h could be interpreted in a similar way although the
correlation here (at the canopy scale) is far weaker.





The weakness of the observed correlations and variability of the observed fluxes are a
reflection of the complexity of in-canopy processes and interactions, all of which (emissions,
photo-chemical production and loss, and turbulent exchange) are strongly influenced by
temperature while only photolysis and direct foliage emissions are directly dependent on light
levels (although the penetration of radiation into the canopy drives both leaf temperature and
turbulence).
## 3.2   Baseline
When FORCAsT is driven in default mode with average meteorology and initial conditions
for July 2012 and only primary emissions of isoprene and monoterpenes, the model fails to
capture either the magnitude or diurnal profile of the observed concentrations and fluxes of
methanol and acetaldehyde at 29 m (Fig. 4(a)-(d); black lines). For both methanol and
acetaldehyde FORCAsT simulates negative fluxes at all times, with a pronounced decrease
during daylight hours (Fig. 4(a) and (c)). Fluxes measured by eddy covariance by contrast
show strongly positive (upward) exchange occurring during the day and fluxes near zero at
night. Observed concentrations increase to 12.8 ppbv (methanol) and 0.72 ppbv
(acetaldehyde) during daylight hours, dipping sharply after dusk and decreasing steadily to a
minimum around dawn (Fig. 4(b) and (d)). The baseline modelled concentrations of both
compounds decrease throughout the 24-hour period, with a dip soon after dawn, and a slight
increase during the early afternoon (Fig. 4(b) and (d)), the latter most likely a result of an
increase in net chemical production. The measurements indicate strong daytime sources of
both methanol and acetaldehyde within the canopy, which FORCAsT does not simulate with
the default model settings.
## 3.3   Biogenic emissions of methanol and acetaldehyde ('E-' simulations)
The pronounced diurnal profile of the observed methanol fluxes with a midday peak is
strongly reminiscent of light and temperature dependent biogenic emissions similar to
isoprene. Leaf-level measurements of methanol emissions have demonstrated that all $C_3$
vegetation types emit methanol at rates on a par with the major terpenoids (Fall and Benson,
1997). Given the lack of other in-situ sources of methanol, the diel cycle of fluxes and
concentrations which is generally absent from anthropogenic and transported sources, and the
magnitude of the underestimation of canopy-top fluxes (ranging from ~0.01 overnight to 0.7





mg m$^{-2}$ h$^{-1}$ in the early afternoon), it seems likely that there are substantial foliage emissions
of methanol at Harvard Forest (see also McKinney et al., 2011).
While the magnitude of the missing acetaldehyde fluxes is lower (between ~0.01 and 0.05 mg
m$^{-2}$ h$^{-1}$), the diel cycles of both fluxes and concentrations is similar to those of methanol. This
again suggests relatively strong leaf-level emissions of acetaldehyde at this site. It is likely
that the absolute concentrations and fluxes are lower as primary emissions of acetaldehyde
have generally been found to be a factor of 2-10 lower than those of methanol (Seco et al.,
2007; Karl et al., 2003; Guenther et al., 2012) and acetaldehyde has chemical sources and
sinks that are relevant at the timescale of canopy exchange.
Fig. 5 shows the relative contributions of the competing processes driving the evolution of
methanol and acetaldehyde within and just above the canopy over the course of the day for
the E-combo90 and E-combo simulations respectively. Concentrations of both oVOC (Fig. 5a
and 3g) increase strongly at all levels from a minimum around dawn. In the case of methanol
(Fig. 5a) there is a clear maximum just below the top of the canopy corresponding to the most
densely foliated level where emissions also peak. This feature is less evident in the case of
acetaldehyde (Fig. 5g) demonstrating its greater number of sources and sinks. Chemical
production and loss is highest at the top of the canopy and the boundary layer just above due
to the higher levels of radiation and temperature driving OH radical formation and reaction
rates. For both oVOC it is emissions and deposition, both leaf-level processes governed by the
stomata, that dominate production and loss; chemistry contributions are at least an order of
magnitude lower. However, both chemistry and turbulent transport contribute to the
complexity evident in the evolution of concentrations and fluxes and the high degree of
variability seen in the observations (see e.g. Figs. 3 and 5).
Difficulties in simultaneously reconciling both fluxes and concentrations of methanol and
acetaldehyde are also likely a result of the complexity of in-canopy processes. Fig. 5 shows
that the top of the canopy is a region of abrupt transition for the sources and sinks of oVOC
with emissions and deposition limited to the canopy and a sudden change in turbulent mixing
above the foliage. The instantaneous nature of concentrations, concentration gradients and
fluxes of methanol and acetaldehyde in time and space are evident from Fig. 5 which
demonstrates that the level at which model and measurements are compared can also affect
the measured-modeled bias, an effect compounded by the instantaneous nature of the model
output fluxes (Eq. 1).



### 3.3.1 Methanol

The effect of introducing the different mechanisms of methanol emissions (simulations E-direct, E-storage, E-combo; Table 5) on fluxes and concentrations of methanol are shown in Fig. 4(a) and (b). Storage emissions are dependent only on temperature and therefore remain relatively high overnight. While modelled fluxes of methanol are positive when storage emissions are included and peak during the middle of the day, modelled midday fluxes are only around a third of measured fluxes (Fig. 6(a); E-storage) and modelled night-time fluxes are well above (~0.15-0.20 mg m$^{-2}$ h$^{-1}$) those observed which are close to but slightly below zero. The diurnal profile of E-storage modelled concentrations is the inverse of measured methanol mixing ratios: elevated at night and decreasing toward the middle of the day (Fig. 6(b); E-storage). This gives further credence to the light-dependent nature of methanol emissions, which has been identified at numerous other forest ecosystems (see e.g. Wohlfahrt et al., 2015; Seco et al., 2015; McKinney et al., 2011).

Direct emissions by contrast are intrinsically linked to photosynthesis and are therefore strongly dependent on light as well as temperature. Introducing purely direct emissions of methanol in FORCAsT (E-direct) reproduces the observed diurnal profile of both fluxes and concentrations and succeeds in capturing the pronounced daytime peak and sharp drop-off at night seen in both. Modelled mixing ratios, however, peak slightly in advance of the observed maximum (Fig. 6(b); E-direct) and do not drop sharply enough after dusk. Modelled fluxes remain negative at night (Fig. 4(a); E-direct) but are slightly below those observed during the dawn transition period, suggesting that while methanol emissions are light dependent they may not be purely direct emissions (which drop to zero at night), although the limitations of eddy covariance flux measurement techniques at night may introduce error into the observation-model comparison.

Combo emissions comprising 80% direct and 20% storage emissions (E-combo) do not reproduce the observed decrease in fluxes and concentrations at night. Modelled nighttime fluxes remain positive and ~0.05-0.1 mg m$^{-2}$ h$^{-1}$ above those observed (Fig. 6(a); E-combo), although as noted above, nighttime flux measurements usually have the greatest uncertainties due to the potential for stable boundary layers and changes in the flux footprint. Additionally, modelled concentrations do not rise sufficiently during the day (with a maximum discrepancy of ~1.5-2 ppbv or 15%) nor drop as steeply as observations after dusk (Fig. 4(b); E-combo). Increasing the proportion of direct emissions to 90% (Fig. 4(a) and (b)) improves the fit of



both fluxes and concentrations at all times with maximum daytime differences reduced to 0.2
mg m$^{-2}$ h$^{-1}$ (~30%) and 1.0 ppbv (~8%) respectively. Modelled concentrations still fail to
capture the pronounced changes observed at dawn, although this may be the result of
boundary layer dilution and canopy flushing. The E-direct simulation gives the best overall
model-measurement fit of the emissions sensitivity tests, emphasizing the strong light-
dependence of methanol emissions previously noted. Including direct emissions in FORCAsT
simulates the bi-directional fluxes and a diel cycle of concentrations similar to those observed
at this site although does not fully capture all of the features of the field data at times of
transition in particular. However, it should be noted that the fluxes especially represent
instantaneous assessments of a situation that rapidly fluctuates in both time and space, which
may in part account for the discrepancies between model and measurements. In spite of this
caveat, our results indicate that methanol emissions are strongly light-dependent, but that
traditional models of primary biogenic emissions (e.g. MEGAN; Guenther et al., 2012) may
not fully account for the fundamental processes driving methanol exchange between the
canopy and atmosphere even when a small contribution from storage pools (e.g. E-combo90)
is included in the model.

### 3.3.2 Acetaldehyde

Similar to methanol, introducing storage only emissions of acetaldehyde does not capture the
peak in fluxes during the day (Fig. 4(c); E-storage), suggesting that acetaldehyde emissions
are also light dependent. Modelled concentrations are close to those observed during daylight
hours in both magnitude and profile with a maximum difference of ~0.2 ppbv (15%), but do
not reproduce the observed drop in concentration just after dusk nor the rapid increase after
dawn (Fig. 4(d); E-storage). However, the greater complexity of acetaldehyde production and
loss on the timescales involved in canopy-atmosphere exchange makes interpretation of the
concentrations more difficult.
Introducing purely direct emissions of acetaldehyde (E-direct) has the same effect as for
methanol. Fluxes are strongly negative at night in FORCAsT (around 0.01-0.015 mg m$^{-2}$ h$^{-1}$
below observed fluxes – Fig. 4(c); E-direct) and concentrations rise too quickly during the
day, peaking around 4 hours earlier and ~0.10 ppbv (~ 15%) higher than measured mixing
ratios (Fig. 4(d); E-direct) with a maximum over-estimation of ~0.15 ppbv (~25%). The steep
nighttime drop in observed fluxes and concentrations is reflected (although over-estimated) in
the model, but overall the simulations suggest acetaldehyde emissions are not purely direct.



In contrast to methanol, acetaldehyde fluxes are reasonably represented by the inclusion of combo emissions comprising 80% direct emissions (Fig. 4(c); E-combo). This captures the diurnal profile of the observations, although not the midday peak, and does not exhibit the same variability in fluxes around dawn and dusk (which may be attributable to the previously described limitations of eddy covariance at these times). Modelled concentrations are within ~0.01 ppbv of those observed during daylight hours, and drop quickly after dusk (Fig. 4(d); E-combo). When the proportion of direct emissions is increased to 90%, concentrations peak in the late afternoon when measured mixing ratios decline (Fig. 4(d); E-combo90). The maximum discrepancy is around half that of E-direct and the nighttime decrease in mixing ratios is well captured. Daytime fluxes are similar to those of the E-combo simulation but decrease more sharply in the afternoon and are lower overnight (~0.05 mg m$^{-2}$ h$^{-1}$ below observations). None of the simulations captures the observed dip in concentration in the late afternoon. The results suggest that the canopy-atmosphere exchange of acetaldehyde may be best represented using the combination of emissions of traditional emissions models, with the "light-dependent" fraction of 80% as currently suggested (Guenther et al., 2012).

### 3.4 Effect of stomatal conductance on modelled emissions (S- simulations)

Because direct emissions use only PAR to explain the diurnal cycle of direct emissions (e.g., the E-direct simulations), here we test the effects of stomatal control on the storage-based emissions mechanism by including stomatal regulation in the storage and combo emissions algorithms. These S- simulations effectively introduce a degree of light-dependence to releases of VOCs from storage pools, although it should be noted that the dependence on PAR introduced in this way is not as strong as for direct emissions. We first present and discuss the results of incorporating stomatal control throughout the day (i.e. the S- simulations using $R_{fct}$ as shown in Eq. 10) for both methanol and acetaldehyde. The effects of modifying the control factor (i.e. the R- simulations using $R_{fct}$ as shown in Eqs. 11a and 11b) are described in Section 3.4.

### 3.4.1 Methanol

The inclusion of stomatal control of methanol emissions from storage structures into FORCAsT improves the fit of modelled to observed fluxes of methanol for both emissions scenarios that include storage-type emissions (Fig. 6a). For 100% storage emissions, stomatal control (Fig. 6a; S-storage *vs.* E-storage), daytime fluxes are enhanced and exhibit the





pronounced midday peak of the measurements; peak modelled fluxes are now generally <0.2
mg m$^{-2}$ h$^{-1}$ below those observed. Night-time fluxes are reduced by ~0.1-0.15 mg m$^{-2}$ h$^{-1}$
bringing them much closer to observations. However, modelled fluxes are still positive at all
times, whereas negative fluxes were measured overnight at the tower. Modelled methanol
concentrations now show a slight post-dawn dip followed by a rapid increase, reaching a
plateau around 11:00 EST. At this point modelled concentrations diverge from those observed
which continue to rise steeply until dusk, peaking nearly 2.5 ppbv (~ 25%) above modelled
levels which rise little during the day. Modelled concentrations continue to remain relatively
steady while observed concentrations drop off sharply at night (Fig. 6b; S-storage), indicating
a dependence on light that is not adequately represented by including stomatal control.
However, some of this behaviour may be a reflection of the discontinuities in modelled
stomatal resistance evident at very low values of PAR (Fig. 2a; just after dawn and before
dusk). This was tested in a later set of sensitivity experiments in which these discontinuities
were smoothed by increasing the level of PAR taken as "daylight". While this had a
substantial effect on modelled $R_{stom}$ in the lower canopy levels following dawn and preceding
dusk (Fig. 2h), the impact on $R_{fct}$ and hence simulated emissions was small. The effect on
fluxes and concentrations was limited to early morning and late afternoon and was negligible
even then (maximum changes of <10% in fluxes and <5% in concentrations at any time; not
shown).
Modelled fluxes for combo emissions (20% storage emissions) with stomatal control (Fig. 6a;
S-combo) mirror those for S-storage although remain slightly higher during the middle of the
day and drop a little closer to zero at night. The burst of methanol escaping the canopy just
after dusk is less pronounced in S-combo. S-combo fluxes are close to those simulated by
increasing the proportion of direct emissions to 90% (E-combo90) apart from the period
preceding dusk when stomatal control acts to reduce fluxes sharply. S-combo concentrations
of methanol in FORCAsT are also similar to the S-storage simulation but continue to rise
until around 16:00 EST at which point they are <0.5 ppbv below measured mixing ratios (Fig.
6b; S-combo). However, the diurnal profiles of methanol concentrations simulated by combo
emissions without stomatal control (E-combo and E-combo90) is closer to the observed, with
90% direct (E-combo90) providing a better overall fit than either of the simulations
incorporating stomatal control.





### 3.4.2 Acetaldehyde

The effects of including stomatal control of emissions of acetaldehyde from storage pools (Fig. 6c and d) are similar to those described above for methanol. For 100% storage (S-storage *vs.* E-storage) emissions the diurnal profile of modelled acetaldehyde fluxes is a good fit to observations (Fig. 6c) with a pronounced peak during the middle of the day (~0.005-0.01 mg m$^{-2}$ h$^{-1}$ (maximum 0.03) below measured fluxes) and dropping below zero overnight (again ~0.005-0.01 mg m$^{-2}$ h$^{-1}$ below measurements). Modelled fluxes also show a short-lived dip at ~16:00 EST which does not appear to be seen in the measurements although the high scattering of observations at either end of the day make it difficult to be certain. S-storage modelled concentrations rapidly increase during the day, peaking ~0.15 ppbv (~25%) above those observed and ~4 hours earlier (Fig. 6d). After this peak the model exhibits the night-time decrease in concentrations seen in the observations (Fig. 6d) but still fails to fully reproduce the post-dawn minimum.

Model output for the S-combo simulation is almost identical to that for S-storage described above, with the two diverging only at night when the combo emissions are lower reducing fluxes and, to a lesser extent, concentrations of acetaldehyde. Although introducing stomatal control of emissions from storage pools improves the magnitude and diurnal profile of modelled fluxes, particularly for the 100% storage emissions pathway (S-storage), acetaldehyde exchanges at Harvard Forest do not show a strong dependence on stomatal conductance at the canopy scale. Instead they are better represented by the use of traditional emissions models with a proportion of emissions from storage pools and the remainder via direct release (with the best fit given by 80% direct and 20% storage, i.e. E-combo). This is in agreement with the theoretical conclusions reached by Niinemets and Reichstein (2003b) and experimental and field results from Kesselmeier (2001) and Kesselmeier et al. (1997). Jardine et al. (2008) report strong evidence of stomatal control at the leaf and branch level and present field measurements that appear to demonstrate that stomatal regulation is relevant at the ecosystem scale for forests in the USA. Our results do not support this conclusion but the authors also reported large differences in the effect of stomatal aperture between tree species (Jardine et al., 2008) which may help explain the apparent contradiction.





### 3.5 Threshold stomatal control (R- simulations)

In the R- simulations, the stomatal control function was modified to limit stomatal regulation of storage emissions to transition periods as outlined in Section 2.3.3. This is consistent with laboratory-based observations of transient emissions bursts associated with light-dark transitions. Furthermore, although stomatal regulation at these times may result in changes to emissions later in the day (as the storage pools have built up while emissions were limited by the stomatal aperture (e.g. Jardine et al., 2008)) the traditional emissions algorithms used in FORCAsT were developed from measured dependencies of emissions on light and temperature and can be assumed to capture actual emission rates well during the middle of the day when the stomata are fully open, even if failing to account for the cause. In these simulations, therefore, we effectively assume that there is a point at which the stomatal aperture is sufficient to no longer be a limiting factor. After this point, the stomatal control factor is set to unity to ensure that emissions are no longer dependent on stomatal aperture. Differences between emissions, and therefore (to a lesser extent) fluxes and concentrations, modelled in the R- and E- simulations should therefore be restricted to the periods around dawn and dusk.

### 3.5.1 Methanol

For 100% storage emissions (R-storage), methanol fluxes show a dip just after dawn and again in the late afternoon, reflecting the period of time when the stomata are partially open (Fig. 7a). At all other times, modelled fluxes match those from the E-storage simulation as expected. Observations do show a drop in methanol fluxes at dawn (~1-2 hours earlier than modelled and greater in magnitude) but not at dusk and are negative overnight. R-storage methanol concentrations match neither the magnitude nor diurnal profile exhibited by the measurements, decreasing during the day as per E-storage but taking longer to recover in the late afternoon (Fig. 7b). The pattern is similar for E- and R-combo emissions with differences between the simulations limited to the transition periods though the effect is less pronounced than the 100% storage case. Fluxes and concentrations are close to those modelled by the E-combo emissions pathway throughout the simulation period, diverging only around dawn and dusk when the stomatal control simulation (R-combo) drops below that without stomatal regulation (E-combo). However, methanol fluxes and concentrations measured above the canopy at Harvard Forest are still most closely matched with the E-direct emissions pathway (Fig. 7a,b).





### 3.5.2 Acetaldehyde

By contrast, acetaldehyde fluxes for the R-storage simulation show very little change from E-storage until late morning (Fig. 7c). During the middle of the day, R-storage fluxes are nearly double those modelled in E-storage but remain well below those observed. Following a steep decline in fluxes in the afternoon to a minimum just before dusk, the post-dusk spike in fluxes previously noted in the 100% storage emissions simulations is enhanced. Acetaldehyde concentrations for R-storage differ little from E-storage during the day but remain elevated at night (Fig. 7d). Introducing stomatal regulation to combo emissions (R-combo *vs.* E-combo) has little effect on either fluxes or concentrations with fluxes exhibiting a distinct minimum at dusk (Fig. 7c) and concentrations dropping slightly earlier in the afternoon (Fig. 7d). Observed acetaldehyde fluxes and concentrations are still best reflected with combo emissions and are well captured with the E-combo "traditional" emissions algorithms without explicit parameterisation of stomatal regulation.

### 3.6 Scaling factor, *n*

The temporally limited effect of stomatal control in our model simulations is consistent with conclusions drawn from a theoretical study based on results from detailed laboratory experiments (Niinemets and Reichstein, 2003b; Niinemets and Reichstein, 2003a), showing that the stomatal control of biogenic VOC emission rates occur over short timescales. These results suggest that regulation of emissions by stomata occurs over too brief a period to be of significance at an ecosystem scale for highly volatile VOCs. On the other hand, they postulate that emission rates of those VOCs such as methanol that are highly soluble in water and therefore have a high liquid-phase concentration are subject to regulation by stomatal conductance over longer timescales, potentially modifying emissions over scales relevant to other processes involved in canopy-atmosphere exchange. Although this does not appear to be the case at Harvard Forest as the initial introduction of stomatal regulation of emissions into FORCAsT does not modify fluxes over an extended period (Fig. 7), additional simulations were conducted to further explore the hypothesis put forward by Niinemets and Reichstein (2003b). In that analysis the authors concluded that the strength and persistence of stomatal control on leaf-level emissions depended strongly on the solubility of the emitted compound, with more soluble molecules (Henry's Law coefficient $H$ less than 100 Pa m$^3$ mol$^{-1}$) more affected. Furthermore they showed that for compounds with $H$ on the order of $10^{-2}$-$10^1$ Pa m$^3$





$mol^{-1}$, the degree to which emissions were influenced by the stomata scaled with $H$. Both
acetaldehyde and methanol have solubilities in this range. Hence in the final stomatal control
simulations (R-storageN15, R-storageN6, R-comboN15 and R-combo6) we scaled the
"degree" of regulation by altering the scaling factor, $n$, in Eqs. 11a and 11b (see Table 5). In
addition to scaling $R_{fct}$ this also alters the duration of stomatal control (i.e. the time taken for
$R_{fct}$ in Eq. 10 to reach values over 1.0) as shown in Fig. 2. Reducing $n$ reduces the time over
which stomatal control regulates emissions (Fig. 2d and e). Doubling $n$ increases the duration
of stomatal control to such an extent that emissions from the lower canopy levels are limited
by the stomata throughout the day (Fig. 2f and g).
In the case of methanol, changing $n$ makes little difference to simulated daytime fluxes (Fig.
7a; R-storageN6 and R-comboN6).  However, night-time fluxes were enhanced slightly
($\sim$0.02 mg m$^{-2}$ h$^{-1}$ for 100% storage emissions and $\sim$0.01 mg m$^{-2}$ h$^{-1}$ for 80% storage
emissions) when $n$ was doubled. Night-time concentrations (Fig. 7b) also increased slightly
around dawn particularly for R-storageN6 ($\sim$0.4 ppbv above R-storage mixing ratios). In
contrast, concentrations at Harvard Forest were observed to fall sharply in the early morning.
Modelled mixing ratios were reduced slightly throughout the day when $n$ was increased in
both storage and combo simulations (R-storageN6 and R-comboN6) with strong decreases
followed by rapid recovery in the late afternoon. Changes at all times were negligible when $n$
was reduced to 1.5 (not shown).
The effects on acetaldehyde concentrations and fluxes are similar, with increases in both at
night when $n$ is increased to 6, although acetaldehyde fluxes are also slightly enhanced during
the middle of the day ($\sim$0.002 mg m$^{-2}$ h$^{-1}$ for both R-storageN6 and R-comboN6; Fig. 7c).
Changes in daytime concentrations are limited to the times around dawn and dusk (Fig. 7d).
Halving $n$ also has little effect on acetaldehyde (not shown).
These results are consistent with observations of canopy structure at Harvard Forest; foliage is
densest in the upper canopy. Fig. 2 shows that changing $n$ has the biggest impact on the lower
canopy levels where light is limited, foliage biomass is low (over 50% of the biomass is found
in the top 20% of the canopy at Harvard Forest; Parker (1998)) and emission rates small.
Our simulations again suggest that the complexity of the competing in-canopy processes act
to buffer the stomatal control of emissions observed at the leaf and branch level. Stomatal
aperture appears to affect emissions over too short a timescale to be observable at the canopy
scale when other sources and sinks are fully accounted for. The times around dawn and dusk



are also associated with rapid changes in chemistry and atmospheric dynamics, which likely outweigh the small differences in emission rates due to stomatal control. Our findings indicate that the inclusion of a "light-dependent fraction" in current emissions algorithms capture the changes in storage emissions due to changes in stomatal aperture sufficiently well at the canopy-scale.

## 4    Conclusions

When light-dependent emissions of methanol and acetaldehyde were included, the FORCAsT canopy-atmosphere exchange model successfully simulated the bi-directional exchange of methanol and acetaldehyde at Harvard Forest, a northern mid-latitude mixed deciduous woodland. The light-dependence of both methanol and acetaldehyde emissions at the leaf-level has been ascribed to the stomatal control of diffusion from storage pools, which would otherwise be expected to be dependent on temperature alone. We incorporated a simple parameterisation of the regulation of emissions according to stomatal aperture into FORCAsT to determine how stomatal control affects canopy-top fluxes and concentrations of methanol and acetaldehyde at this site.

We found that although some simulations that included stomatal regulation of emissions showed a good fit to measured fluxes, none proved effective in reproducing both the observed concentrations and fluxes. Instead, our simulations show that the algorithms currently used for modelling foliage emissions of oVOC are capable of capturing fluxes and concentrations of both methanol and acetaldehyde near the top of the canopy and are therefore appropriate for use at the ecosystem-scale. Our results demonstrate that canopy-top fluxes of methanol and acetaldehyde are determined primarily by the relative strengths of foliage emissions and dry deposition indicating that 3-D atmospheric chemistry and transport models must include a treatment of deposition that is not only dynamically intrinsically linked to land surface processes but is consistent with the emissions scheme.

Our results show that it is possible to model canopy top fluxes of methanol and acetaldehyde, and to capture bi-directional exchange without the need for including direct representations of stomatal control of emissions. Overall, we find that the bi-directional exchange of methanol at Harvard Forest is best modelled with traditional emissions models (e.g. MEGAN; Guenther et al. 2012) assuming 100% light-dependent (direct) emissions. In the case of acetaldehyde, modelled concentrations prove robust with a relatively good fit to observations for all emissions scenarios employed here, likely due to the greater number of chemical sources and





sinks of acetaldehyde in comparison to methanol. Canopy-top acetaldehyde fluxes at this site
are also best modelled with emissions algorithms that include light dependence directly (i.e.
without the introduction of stomatal control). In contrast to methanol, however, acetaldehyde
emissions at Harvard Forest appear to be derived from both direct synthesis and storage pools,
with 80% direct emissions giving the best overall fit.
Given that observed methanol fluxes appear strongly correlated with stomatal conductance at
small stomatal apertures it is perhaps surprising that we found no evidence supporting the
suggestion that stomatal control of methanol emissions are observable at the canopy scale. It
is likely therefore that this correlation is ascribable to the strong dependence of methanol
deposition on stomatal resistance.
Our results highlight the importance of the holistic treatment and coupling between land
surface sources and sinks. The use of explicit and consistent dynamic representations of
emissions and deposition, which dominate the in-canopy budgets for these longer-lived
oVOC, are needed in atmospheric chemistry and transport models. Such an approach would
adequately account for the role of the stomata in both processes and allow bi-directional
exchange to be successfully simulated without the need for including either leaf-level process
detail or a compensation point.
However, this study also demonstrates the need for a better understanding and representation
of the complex relationship between turbulence, fluxes and concentration gradients within and
above the forest canopy. Such understanding can only be achieved through further modelling
studies at a range of scales in combination with robust measurements of concentrations and
fluxes of VOCs, their primary oxidants and oxidation products at multiple heights within the
forest canopy.
**Acknowledgements**
This material is based upon work supported by the National Science Foundation under Grant
No. AGS 1242203.





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



Table 1. Atmospheric and meteorological measurements relevant to this study made between
7[th] June and 24[th] September 2012 at the EMS Tower in Harvard Forest.

| Type | Measurement | Height (m) | Instrument |
|---|---|---|---|
| **Chemical** | | | |
| Methanol, $CH_3OH$[a] | Concentration, Flux | 29 | PTR-TOF-MS, Iconicon Analytik |
| Acetaldehyde, $CH_3CHO$[a] | Concentration, Flux | 29 | PTR-TOF-MS, Iconicon Analytik |
| $CO$[b] | Concentration | 29 | Modified IR-absorption gas-filter correlation analyser |
| $O_3$[b] | Concentration | 29, 24.1, 18.3, 12.7, 7.5, 4.5, 0.8, 0.3 | UV absorbance instrument |
| Water Vapour[c] | Concentration | 29 | Licor $CO_2$-$H_2O$ sensor |
| **Meteorological** | | | |
| Air temperature[c] | | 29, 27.9, 22.6, 15.4, 7.6, 2.5 | 30kW precision thermistor in aspirated radiation shield |
| PAR[c] | | 29, 12.7 | Quantum sensor |
| Windspeed[c] | Horizontal, vertical | 29 | AT1 sonic anemometer |
| Wind direction[c] | | 29 | AT1 sonic anemometer |
| Relative humidity[c] | | 29, 22.6, 15.4, 7.6, 2.5 | Thin film capacitor sensor in aspirated radiation shield |

[a]data provided by McKinney and Liu; [b]Munger and Wofsy (1999b); [c]Munger and Wofsy (1999a)
Table 2. Boundary and initial conditions used for the FORCAsT simulations.

| Model parameter or variable | Value |
|---|---|
| Total leaf area index ($m^2$ leaf area $m^{-2}$ ground area)[a] | 3.67 |
| Average canopy height (m)[b] | 23.0 |





| | |
|---|---|
| Average trunk height (m)[b] | 6.0 |
| **Meteorology (values measured at 29 m)** | |
| Air temperature (°C)[c] | 20.9 |
| Wind speed (m s$^{-1}$)[c] | 1.589 |
| Friction velocity, u$^{*}$ (m s$^{-1}$)[d] | 0.278 |
| Standard deviation of vertical wind velocity, $\sigma_w$ (m s$^{-1}$)[d] | 0.351 |
| **Concentrations at 29 m (ppbv)** | |
| Isoprene[e] | 0.939 |
| Total monoterpenes[e] | 0.449 |
| MVK-MCR[e] | 0.786 |
| Methanol[e] | 10.11 |
| Acetaldehyde[e] | 0.620 |
| Acetone[e] | 2.608 |
| Ozone[f] | 33.54 |
| CO[f] | 164.8 |
| Water vapour[c] | 1.861% |
| **Miscellaneous** | |
| Ozone at ground-level (0.3 m)[f] | 20.35 ppbv |
| Temperature at ground-level (2.5 m)[c] | 18.1 °C |
| Soil Temperature at 15, 40, 50 and 90 cm depth[a] | 24.9, 25.9, 25.9, 21.4 °C |
| Soil Moisture at 15, 40, 50 and 90 cm depth[a] | 0.18, 0.15, 0.17, 0.18 |
| NO$_2$ at 29 m[g] | 1.00 ppbv |
| N$_2$O$_5$ at 29 m[g] | 1.50 ppbv |





[a]Munger and Wofsy (1999c); [b]Parker (1998); [c]Munger and Wofsy (1999a); [d]data provided by Munger; [e]data
provided by McKinney and Liu; [f]Munger and Wofsy (1999b); [g]Moody et al. (1998)
Table 3. Deposition parameters for methanol and acetaldehyde.

| Chemical | Henry's Law constant | Diffusivity | Reactivity factor |
|---|---|---|---|
| Methanol | $2.2E02^a$ | $1.33^b$ | $1.0^c$ |
| ALD1 (acetaldehyde)[d] | 11.4 | 1.6 | 1.0 |

[a]Sander (1999); [b]Wesely (1989); [c]Karl et al. (2010); [d]Ashworth et al. (2015)
Table 4. Values of stomatal resistance coefficients and parameters used in FORCAsT.

| Coefficient | Value |
|---|---|
| $r_{smin}$ | 90.0 |
| $b_{rs}$ | 200.0 |
| $T_{min}$ | -2.0 |
| $T_{max}$ | 45.0 |
| $T_0$ | 30.0 |
| $b_v$ | 0.5 |
| $a$ | 0.066667 |
| $b_\varphi$ | 1.6666667 |

Table 5. Modifications to the base case for each of the sensitivity simulations.

| Simulation | Changes from baseline simulation |
|---|---|
| Emissions (E) of methanol and acetaldehyde included as: | |
| **E-direct** | 100% direct emissions |
| **E-storage** | 100% storage emissions |
| **E-combo** | 80% direct; 20% storage |
| **E-combo90** | 90% direct; 10% storage |
| Stomatal control (S) of storage emissions included: | |
| **S-storage** | Activity factor, $\gamma_T$, for storage emissions scaled by stomatal control factor, |





| | |
|---|---|
| | $R_{fct}$ (Eqs. 2 and 9, with $n$=3) |
| **S-combo** | Activity factor, $\gamma_T$, for storage emissions scaled by stomatal control factor, $R_{fct}$ (Eqs. 9 and 10, with $n$=3); 80% direct and 20% storage |

Stomatal control of storage emissions using modified stomatal control factor, $R_{fct}$ (R):

| | |
|---|---|
| **R-storage** | Threshold stomatal control factor used (Eq. 11) |
| **R-storageP** | Threshold stomatal control factor used (Eq. 11) and daytime threshold for PAR increased to 10.0 |
| **R-storageN15** | Threshold stomatal control factor used (Eq. 11) with scaling factor $n$ set to 1.5 |
| **R-storageN6** | Threshold stomatal control factor used (Eq. 11) with scaling factor $n$ set to 6.0 |
| **R-combo** | Threshold stomatal control factor used (Eq. 11); 80% direct and 20% storage |
| **R-comboP** | Threshold stomatal control factor used (Eq. 11) and daytime threshold for PAR increased to 10.0; 80% direct and 20% storage |
| **R-comboN15** | Threshold stomatal control factor used (Eq. 11) with scaling factor $n$ set to 1.5; 80% direct and 20% storage |
| **R-comboN6** | Threshold stomatal control factor used (Eq. 11) with scaling factor $n$ set to 6.0; 80% direct and 20% storage |

1    Table 6a. Emission factors (nmol m$^{-2}$ (projected leaf area) s$^{-1}$) for VOCs included in

2    FORCAsT baseline simulation.

| VOC | Direct | Storage |
|---|---|---|
| Isoprene | 4.83[a] | 0.000 |
| α-pinene | 0.000 | 0.071[b] |
| β-pinene | 0.000 | 0.032[b] |
| *d*-limonene | 0.000 | 0.054[b] |
| Methanol | 0.000 | 0.000 |



| | | | |
|---|---|---|---|
| Acetaldehyde | 0.000 | | 0.000 |

1   [a]Helmig et al. (1999); [b]Geron et al. (2000)

2   Table 6b. Emission factors, ε (nmol m$^{-2}$ (projected leaf area) s$^{-1}$) and total canopy emissions

3   (mg m$^{-2}$ day$^{-1}$) for methanol and acetaldehyde for the FORCAsT simulations in Table 5.

| oVOC | Methanol | | | Acetaldehyde | | |
|---|---|---|---|---|---|---|
| Simulation | Direct ε | Storage ε | Total | Direct ε | Storage ε | Total |
| E-direct | 4.894 | 0.000 | 435.8 | 0.303 | 0.000 | 28.7 |
| E-storage | 0.000 | 0.653 | 457.0 | 0.000 | 0.036 | 28.5 |
| E-combo | 1.670 | 0.418 | 441.2 | 0.112 | 0.027 | 32.0 |
| E-combo90 | 2.815 | 0.296 | 457.8 | 0.175 | 0.019 | 31.6 |
| S-storage | 0.000 | 0.326 | 441.0 | 0.000 | 0.019 | 32.1 |
| S-combo | 1.065 | 0.266 | 454.7 | 0.063 | 0.015 | 31.3 |
| R-storage | 0.000 | 0.653 | 438.6 | 0.000 | 0.040 | 30.5 |
| R-storageN15 | 0.000 | 0.653 | 429.5 | 0.000 | 0.040 | 31.2 |
| R-storageN6 | 0.000 | 0.751 | 445.6 | 0.000 | 0.046 | 30.9 |
| R-combo | 1.670 | 0.418 | 434.0 | 0.112 | 0.027 | 31.5 |
| R-comboN15 | 1.670 | 0.418 | 435.8 | 0.112 | 0.027 | 28.7 |
| R-comboN6 | 1.670 | 0.418 | 457.0 | 0.112 | 0.027 | 28.5 |
| S-storageP | 0.000 | 0.326 | 441.2 | 0.000 | 0.019 | 32.0 |
| S-comboP | 1.065 | 0.266 | 457.8 | 0.063 | 0.015 | 31.6 |
| R-storageP | 0.000 | 0.653 | 441.0 | 0.000 | 0.040 | 32.1 |
| R-comboP | 1.670 | 0.418 | 454.7 | 0.112 | 0.027 | 31.3 |





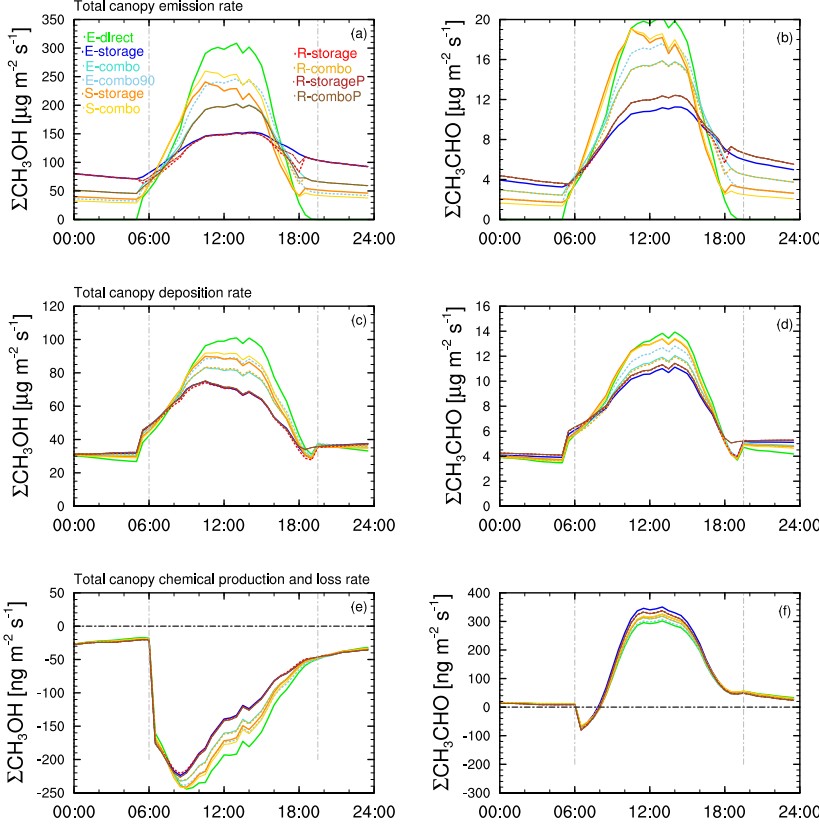

Figure 1. Total canopy production and loss rates per unit ground area for methanol (left) and
acetaldehyde (right) summed over the 10 crown space layers. Coloured lines show total
emissions (top), deposition (middle) and chemical production and loss (bottom) for each
simulation.





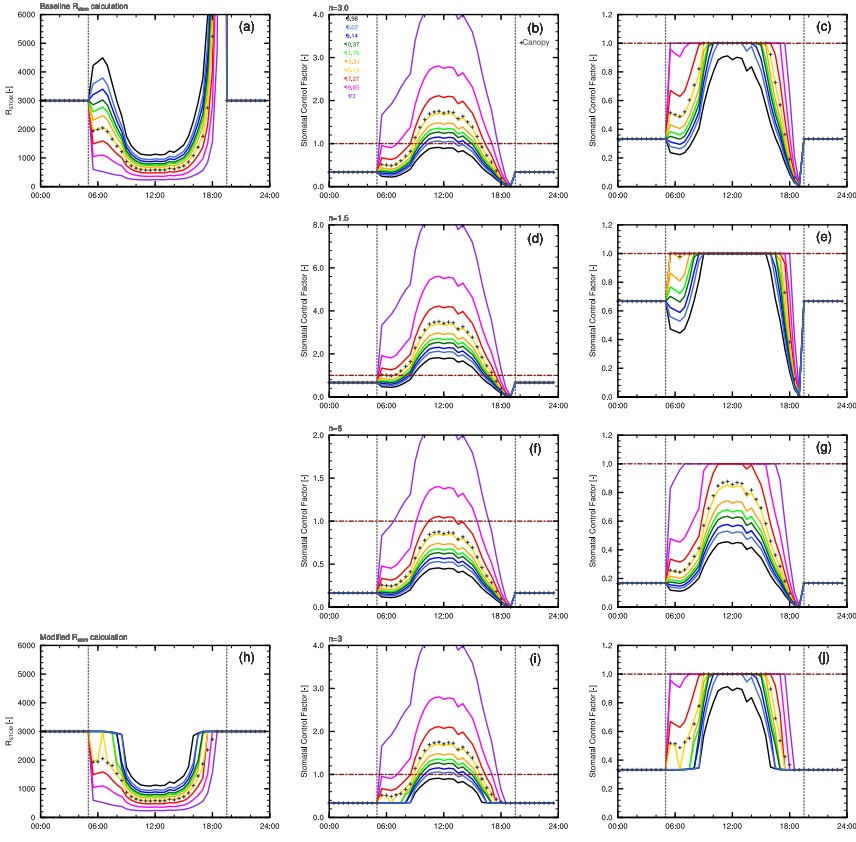

Figure 2. Stomatal control applied to storage emissions. The top row shows the baseline (a)
stomatal resistance, (b) stomatal control factor $R_{fct}$ as calculated in Eq. 10, and (c) the
stomatal control factor as calculated in Eqs. 11a and 11b, i.e. with a limiting value of 1.0.
Coloured lines show the resistances and control factors as a leaf area-weighted average for
each crown space model level across the 10 leaf angle classes. The crosses show the canopy
average weighted by foliage fraction in each level. The second and third rows show the effect
on $R_{fct}$ of altering the scaling factor, $n$, in Eq. 10 ((d) and (f)) and Eqs. 11a and 11b ((e) and
(g)). The bottom row shows the same as the top for the modified stomatal resistance
calculations in which "daylight" is assumed to start only when PAR exceeds a threshold of
10.0 μmol m$^{-2}$ s$^{-1}$.





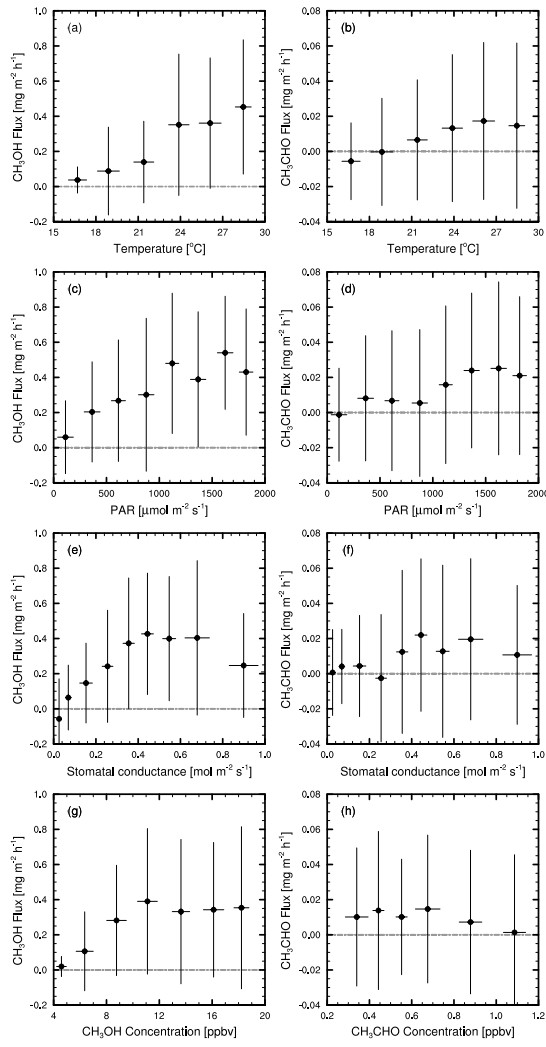

Figure 3. Observed daytime (05:00-19:00 EST) fluxes of methanol (left) for July 2012 versus
(a) air temperature, (c) PAR, (e) canopy stomatal conductance, and (g) methanol
concentration  (all measured at 29 m). The right hand column (panels b, d, f, h) shows the
same relationships for acetaldehyde. Temperatures were binned in 2.5 ºC intervals, PAR in
250 µmol m$^{-2}$ s$^{-1}$, stomatal conductance in ~0.1 mol m$^{-2}$ s$^{-1}$ and concentrations in 2.5 ppbv
increments (methanol) and 0.2 ppbv (acetaldehyde). Average values for each bin are marked
with circles; vertical and horizontal bars indicate 1 standard deviation above and below the
mean in each case.



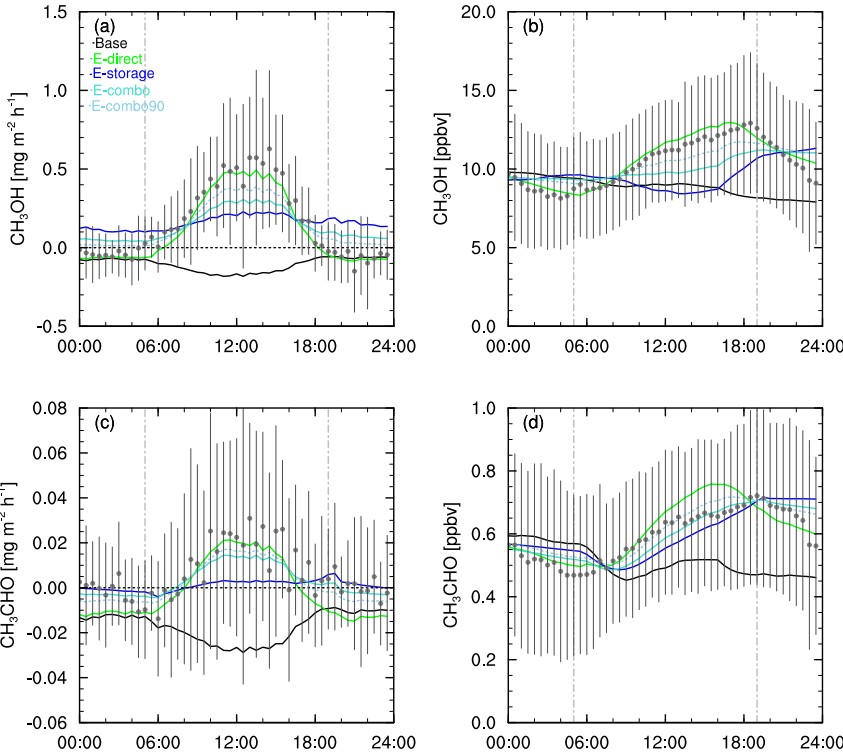

Figure 4. Measured (grey circles with vertical bars indicating 1 standard deviation above and
below the mean) and modelled (solid lines) fluxes (left) and concentrations (right) at 29 m for
an average day in July 2012 for methanol (a) fluxes (mg m$^{-2}$ h$^{-1}$) and (b) concentrations
(ppbv), and acetaldehyde (c) fluxes and (d) concentrations. The solid black line shows the
baseline model simulation. Coloured lines denote E-direct (green), E-storage (blue) and E-
combo (cyan) simulations in which direct, storage and combination emissions pathways
respectively are included. The dashed turquoise line shows the E-combo90 (combo emissions
with 90% direct and 10% storage emission pathways) sensitivity test. Dashed grey vertical
lines show dawn and dusk. Times shown are Eastern Standard Time (EST).





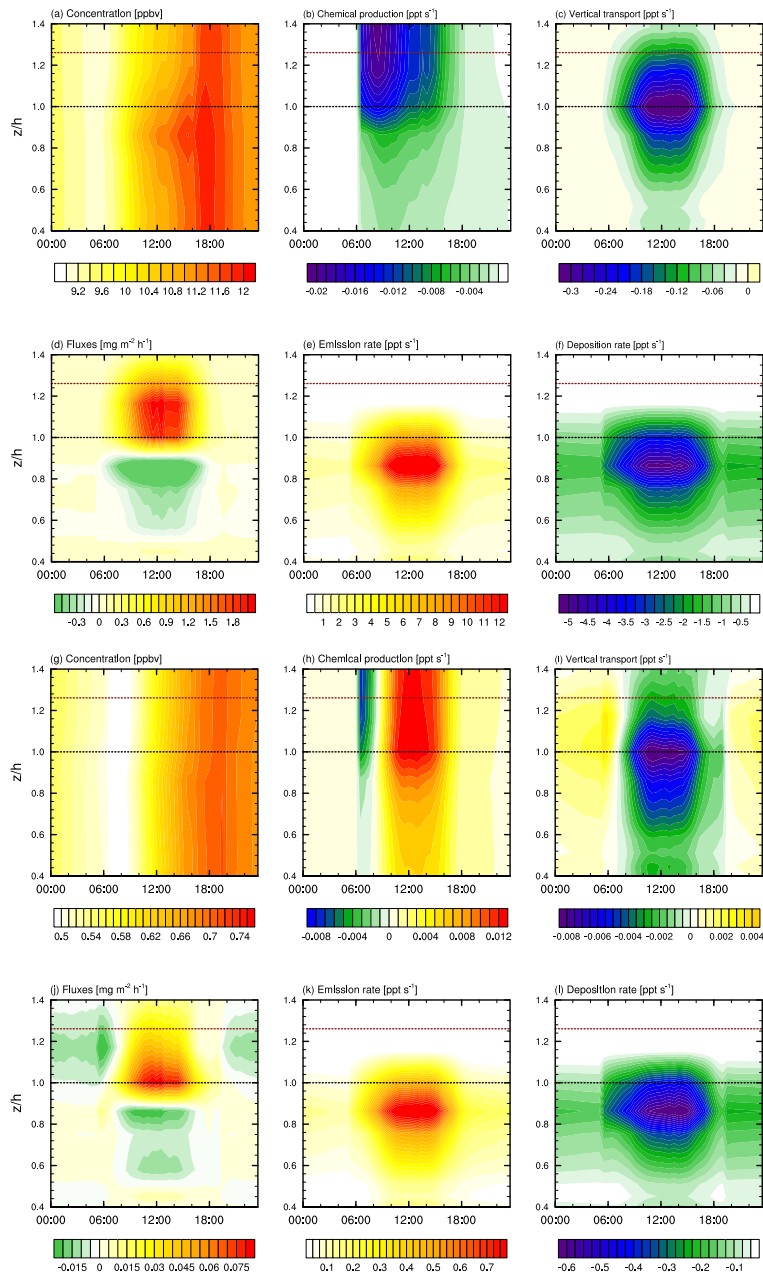

Figure 5. Production and loss within the canopy space for methanol: (a) concentration, (b)
chemical production rate (including photolysis), (c) changes in concentration due to vertical
mixing, (d) flux, (e) emission rates, and (f) deposition rates of methanol for the E-combo90



simulation. Rates are instantaneous in time and space. The vertical axis shows height relative
to canopy top height; times on the horizontal axis are LT. Panels (g)-(l) show the same for
acetaldehyde for the E-combo simulation. Dashed horizontal lines denote canopy top (black)
and observation height (red).

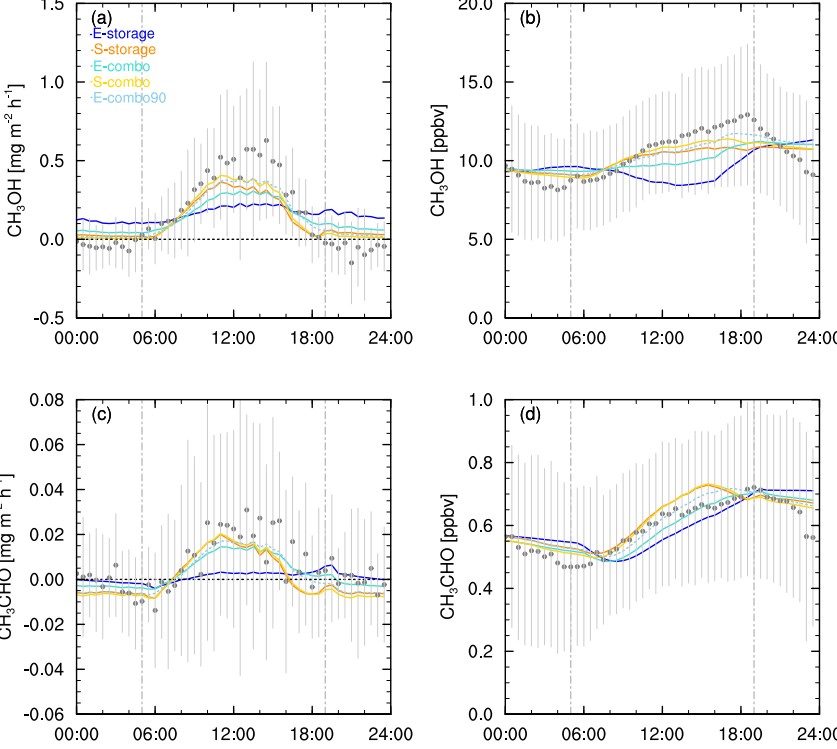

Figure 6. As Fig. 4 with blue lines showing E-storage and orange lines S-storage simulations,
and turquoise and yellow lines showing E-combo and S-combo simulations respectively. The
dashed turquoise line shows the E-combo90 sensitivity test. Panels show (a) methanol fluxes,
(b) methanol concentrations, (c) acetaldehyde fluxes, and (d) acetaldehyde concentrations at
29 m.





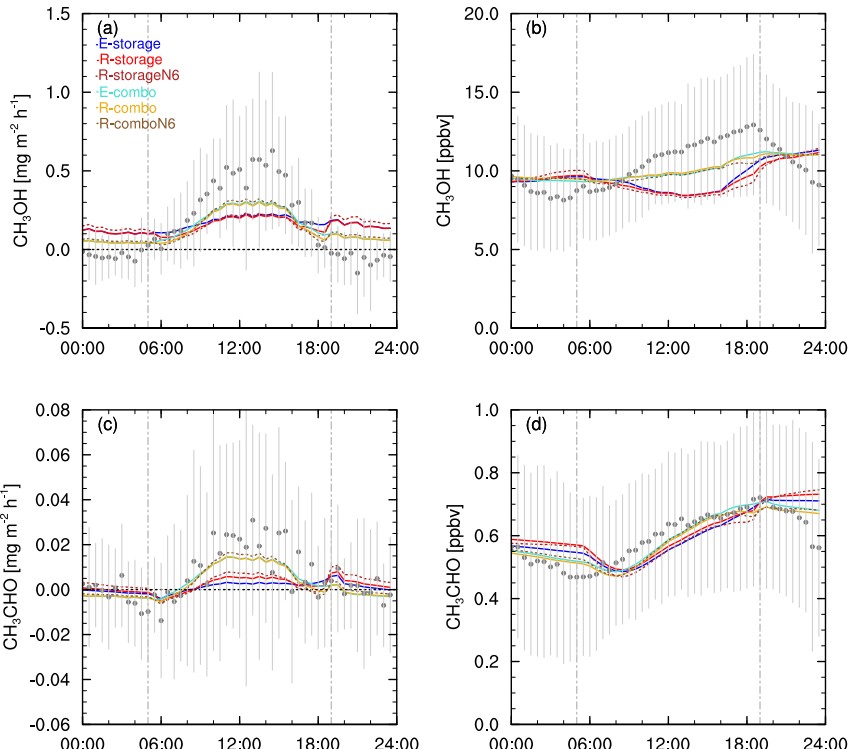

Figure 7. Simulations of modified stomatal control of storage emissions (R-). Blue and
turquoise lines show E-storage and E-combo as Fig. 6. Red (R-storage) and dashed dark red
(R-storageN6) lines show the effects on 100% storage emissions for scaling factor n=3 and
n=6 respectively. Gold (R-combo) and dashed brown (R-comboN6) lines show the same for
combo emissions (20% storage). Panels show (a) methanol fluxes, (b) methanol
concentrations, (c) acetaldehyde fluxes, and (d) acetaldehyde concentrations at 29 m for an
average day in July 2012.

