# Peer review of "Modelling bi-directional fluxes of methanol and"

_Atmospheric Chemistry and Physics, 2016_

## Referee Comment (RC1) · Anonymous Referee #1 · 26 Jul 2016

The authors present a canopy modeling study aimed at better understanding the mechanisms driving emissions and deposition of methanol and acetaldehyde between the forest and atmosphere. They use a 1D canopy model and test the sensitivity of the simulation (and resulting agreement with observations) to various assumptions related to emission pathways (e.g., importance of direct versus storage emissions, and degree of stomatal control over the storage emissions). These are relevant questions with implications for our ability to model these emissions and predict the sensitivity of emissions to environmental changes.

The study is well thought out and carefully executed, with a range of sensitivity tests presented. It merits publication in ACP. Some suggestions are below.

Discussion of all the sensitivity analyses tends to run on a bit to excess. The manuscript could be more effective and concise if some secondary material were moved to a supplement. The "threshold stomatal control" section is one example. Just prior to this, you state that for both methanol and acetaldehyde, stomatal control is not needed to explain the canopy-scale observations. This seems to be one of the main take-home messages and is a useful finding. But once that point is established, it doesn't seem we learn anything substantially beyond that from the section looking at the subtleties of threshold effects. I.e. the take-home from Sections 3.5 and 3.6 seems unchanged from what we had in 3.4; the bulk of this could be moved to SI with a brief summary in the main text.

7L3-6, "While we acknowledge that the magnitudes of the recorded night-time fluxes during summer 2012 may have large associated errors, we are confident in the direction of the exchange as we see variation between different species suggesting no systematic bias." Unclear what this means. Please clarify.

7L30-31, "although its reactions are limited to oxidation by OH to produce formaldehyde". That's the only relevant chemical sink in any case. Perhaps change to "its source/sink reactions..." to clarify that you are not including any chemical sources of methanol (e.g. peroxy radical reactions) ... I concur that these would not be important in this context.

"FORCAsT includes a physical representation of a forest canopy, with the lowest eight model levels set as trunk space and the next ten as crown space. The ten crown space levels contain the foliage". So this neglects any shrubs etc near the ground, is this an ok assumption for Harvard Forest?

10L26-28, "rs" is capitalized in Eqn 3 but not on line 26

Please be more explicit about what assumptions are embedded in the lack of treatment of advection for methanol and acetaldehyde. Both are both sufficiently long-lived that in reality there is a substantial advective component.

13L12-14. Note that acetaldehyde only accounts for 25-40% of the bidirectional VOC flux in the Guenther 2012 scheme, depending on PFT

"The emission factors were then modified to best reconcile modelled and observed concentrations and fluxes at 29 m whilst ensuring that total canopy emissions for all simulations were within $\pm 10\%$." Unclear what this means, please be more specific.

13L25-27. "The greater the contribution from storage the higher the overnight and the lower the daytime peak" Wording is a bit odd. A higher storage flux by itself doesn't necessarily decrease the daytime peak. E.g. if you kept the direct flux constant and increased the storage flux, you'd still have a daytime peak (and it would occur at higher concentration). What you're saying occurs because you've constrained the 24-h integrated canopy emissions to be the same between simulations. Right? Could say "...and the lower the diurnal amplitude".

14L21, where does the limiting nighttime value of 3000 come from?

14L22, what does the scaling factor n physically represent?

Figure 2 is hard to decipher. Font very small. Heights hard to read.

Fig 3, perhaps show the corresponding model correlations that emerge from the simulations?

16L15-32. Figure 3 seems hard to interpret due to convolution between the independent variables. For instance, you state "the data appear to show a strong linear correlation at low conductance, suggesting that at small stomatal aperture the stomata exert control over fluxes of methanol to the extent that it is observable at the canopy scale." But couldn't it equally have nothing to do with stomatal conductance, and just arise from temperature/light affecting both emissions and stomatal conductance simultaneously?

16L21-22. "A similar relationship between canopy-top methanol fluxes and concentrations is likely due to the influence of atmospheric concentrations on dry deposition

to surfaces within the canopy." Similar comment. Temp/light would increase Ch3OH fluxes which would in turn increase the concentrations. And wouldn't your point about deposition work in the opposite direction? At the lowest CH3OH concentration deposition should be lowest so that net emission is highest. But the data go in the opposite direction.

"the level at which model and measurements are compared can also affect the measured-modeled bias, an effect compounded by the instantaneous nature of the model output fluxes". But aren't the model fluxes averaged over the same 30-min intervals as the data?

None of the simulations appear to capture the nighttime concentration decline for methanol. Is this a mixing effect or does it point to some shortcoming in the model treatment of deposition?

Perhaps I missed it, but if not please clarify how atmospheric mixed layer dynamics are treated. Are those entirely prognostic within the model energy balance? How do we know how well the model captures the ML depth and growth/collapse timing, since those would clearly affect the diurnal concentration profiles that you interpret.

Please double- check references.

---

## Referee Comment (RC2) · Anonymous Referee #2 · 21 Sep 2016

Aahworth et al. presented 1D canopy model performance evaluations using an observational dataset at the Harvard forest. The presented model frame work (FORCAsT) successfully captures the observations - bi-directional fluxes of methanol and acetaldehyde. The authors demonstrated improved model performance by adapting light-dependent emission in the model. Overall, the manuscript is very well written, well motivated and informative. As there are not many available 1D canopy model frameworks in the community, this work which presented very thoroughly would become an invaluable addition. I recommend the publication of this discussion paper as it is to ACP.

---

## Author Comment (AC1) · 25 Oct 2016

Author Response to Reviewer #1

We thank the reviewer for their positive comments regarding our study and the manuscript in general, as well as their suggestions for improving our work. Please see our detailed replies below.

*Discussion of all the sensitivity analyses tends to run on a bit to excess. The manuscript could be more effective and concise if some secondary material were moved to a supplement. The "threshold stomatal control" section is one example. Just prior to this, you state that for both methanol and acetaldehyde, stomatal control is

not needed to explain the canopy-scale observations. This seems to be one of the main take-home messages and is a useful finding. But once that point is established, it doesn't seem we learn anything substantially beyond that from the section looking at the subtleties of threshold effects. i.e. the take-home from Sections 3.5 and 3.6 seems unchanged from what we had in 3.4; the bulk of this could be moved to SI with a brief summary in the main text.

We thank the reviewer for this suggestion. We have substantially shortened Section 3.5 and 3.6 to reduce repetition and maintain focus on the take-home messages. We have chosen not to move any material to a supplement as we hope that the edited section addresses the reviewer's concerns. We would agree with the reviewer that this has very much improved the flow of the narrative.

*7L3-6, "While we acknowledge that the magnitudes of the recorded night-time fluxes during summer 2012 may have large associated errors, we are confident in the direction of the exchange as we see variation between different species suggesting no systematic bias." Unclear what this means. Please clarify.

At all times of day, there is variation in flux direction between different species, i.e. they are not all positive or all negative giving us confidence that there is no systematic bias in the Eddy Covariance calculations. Rather the direction of flux is genuinely recording the direction of gradient in concentration across the top of the canopy and is not an artefact of the instrument or methodology. We have re-worded this sentence to explain this more clearly: "While we acknowledge that the magnitudes of the night-time fluxes recorded during summer 2012 may have large associated errors, we are confident that the direction of the exchange is well captured as the observed fluxes for different species were not correlated, suggesting no systematic bias in the application of eddy covariance at this site."

*7L30-31, "although its reactions are limited to oxidation by OH to produce formaldehyde". That's the only relevant chemical sink in any case. Perhaps change to "its

source/sink reactions..." to clarify that you are not including any chemical sources of methanol (e.g. peroxy radical reactions) . . . I concur that these would not be important in this context.

We have changed the wording of this sentence to: "The CACM chemistry mechanism in FORCAsT treats methanol explicitly with no chemical sources (e.g., production from peroxy radicals) and a sink via oxidation by OH to produce formaldehyde."

*"FORCAsT includes a physical representation of a forest canopy, with the lowest eight model levels set as trunk space and the next ten as crown space. The ten crown space levels contain the foliage". So this neglects any shrubs etc. near the ground, is this an ok assumption for Harvard Forest?

There is very little understory vegetation in the vicinity of the EMS Tower at Harvard Forest. Please see attached photos (supplementary zip file) showing the trunk space around the shed at the base of the tower and in the immediate vicinity. In addition, Parker (1998) measured the vertical distribution of the foliage in the canopy and recorded that the understory, i.e. below 6 m (the trunk space height in FORCAsT), accounted for only ∼7.5% of total LAI and received only <10% of top of canopy PAR. Applying allometric relations to a tree inventory taken of the tower footprint in 2010 suggests the understory contains a maximum of ∼8% of the total biomass.

*10L26-28, "rs" is capitalized in Eqn. 3 but not on line 26

Thank you for catching that; we have removed the capitalisation in the equation.

*Please be more explicit about what assumptions are embedded in the lack of treatment of advection for methanol and acetaldehyde. Both are both sufficiently long-lived that in reality there is a substantial advective component.

During July 2012, >60% of air masses arriving at the Harvard Forest site came from north, northwest or west. Lee et al. (2006) showed that concentrations of anthropogenic VOCs were consistently below average under such flow conditions. Furthermore, VOCs advected to the site have been transported over long distances and are generally well-mixed vertically, increasing concentration at all heights and therefore having little impact on the concentration gradient. Over the timescale of our model simulations, the rapid fluctuations due to in-situ production and loss can be expected to dominate over the longer-term, slower changes in advected pollutant concentrations. We conducted a sensitivity test in which acetaldehyde, methanol and acetone were advected at a constant rate just above the top of the canopy and found this slightly dampened the diel cycle of modelled fluxes and concentrations. While some of the observed methanol and acetaldehyde is likely transported to the site from more polluted areas, this result suggests that during the time period of the study the advective contribution is sufficiently minor in comparison to in-situ production. We have added a statement to this effect to the manuscript: "Lee et al. (2006) also reported that air masses reaching the Harvard Forest site form the north, northwest and west had consistently low levels of anthropogenic VOCs. Such conditions prevailed >60% of July 2012 and we found that including advection as an additional source of methanol and acetaldehyde did not improve model fit (results not shown)."

*13L12-14. Note that acetaldehyde only accounts for 25-40% of the bidirectional VOC flux in the Guenther 2012 scheme, depending on PFT

While we used the emission factors developed for MEGANv2.1 (Guenther et al., 2012) as a basis for the factors used here, we adjusted them to optimise the fit between modelled and observed concentrations and fluxes for each simulation. In so doing, acetaldehyde emission factors were ~20-30% of the bidirectional VOC flux in the Guenther et al. (2012) scheme.

It would certainly be of interest to investigate bidirectional fluxes of other species lumped into this group in the Guenther et al. (2012) scheme in the future. However, fluxes of other compounds were found to be at or below instrument detection limits at Harvard Forest in 2012.

"The emission factors were then modified to best reconcile modelled and observed concentrations and fluxes at 29 m whilst ensuring that total canopy emissions for all simulations were within ∼10%." Unclear what this means, please be more specific.

While our baseline emission factors were based on those suggested in Guenther et al., 2012 we constrained the emissions such that there was no more than 10% difference between the total emissions of each species included in each of the simulations. We have re-worded this sentence to read: "The emission factors were then scaled to reconcile modelled and observed concentrations and fluxes at 29 m whilst conserving the total canopy emissions for each species. Twenty-four hour aggregated emissions for each simulation were within ∼10% of each other."

*13L25-27. "The greater the contribution from storage the higher the overnight and the lower the daytime peak" Wording is a bit odd. A higher storage flux by itself doesn't necessarily decrease the daytime peak. E.g. if you kept the direct flux constant and increased the storage flux, you'd still have a daytime peak (and it would occur at higher concentration). What you're saying occurs because you've constrained the 24-h integrated canopy emissions to be the same between simulations. Right? Could say ". . . and the lower the diurnal amplitude".

Correct, the reduction in daytime peak is a result of adjusting the emission factors to maintain similar total emissions between simulations. We have re-worded this sentence as suggested: "The greater the contribution from storage the higher the overnight fluxes and the smaller the diurnal amplitude. . .."

*14L21, where does the limiting nighttime value of 3000 come from? The limiting value is the model default value for nighttime stomatal resistance for calculating deposition rates and includes cuticular resistance. Jarvis (1976) reports a maximum stomatal resistance of 1000. We therefore introduced the scaling factor n, with an initial value of 3, giving 3000/3 (i.e. the Jarvis maximum) as a suitable starting point for our sensitivity tests (see below).

[Figure]

*14L22, what does the scaling factor n physically represent? The scaling factor was introduced to scale the model night-time resistance which includes cuticular resistance as outlined above to the value reported by Jarvis (1976). In combination the two (3000/n) represent the linear dependence of emission rate on stomatal aperture. The parameter values used in the stomatal conductance model (Jarvis, 1976) are species-specific. We therefore conducted a series of sensitivity tests to assess the validity of our chosen starting point. These tests indicated that the precise value of n used in this function did not alter our conclusion that the introduction of explicit stomatal control is not necessary to adequately model emissions and fluxes at the canopy scale.

We have clarified these parameter values by adding the following to the description of the stomatal control algorithm in Section 2.3.3: "... 3000 is the model default limiting night-time value of Rstom and n is a scaling factor. The night-time "stomatal" resistance in fact includes the cuticular resistance and n was introduced to account for this. The value of n was initially set to 3 for the S-storage and S-combo simulations, as Jarvis (1976) reported a limiting value of 1000 although this was species-dependent. The effect of the choice of value of n is explored in Section 3.5."

*Figure 2 is hard to decipher. Font very small. Heights hard to read.

We have increased the font size of the axes titles and scales. The new version is attached.

*Fig 3, perhaps show the corresponding model correlations that emerge from the simulations?

We appreciate this suggestion, but decided against showing model correlations as our focus was on explaining the observations and targeting the processes that the measurements suggested were important in determining the direction of the fluxes at the top of the canopy.

*16L15-32. Figure 3 seems hard to interpret due to convolution between the independent variables. For instance, you state "the data appear to show a strong linear correlation at low conductance, suggesting that at small stomatal aperture the stomata exert control over fluxes of methanol to the extent that it is observable at the canopy scale." But couldn't it equally have nothing to do with stomatal conductance, and just arise from temperature/light affecting both emissions and stomatal conductance simultaneously?

Absolutely. We included Figure 3 to show the reasoning that led us to investigate whether stomatal control was a factor in bi-directional exchange of methanol and acetaldehyde. We agree that the processes are highly coupled and strongly dependent on the same environmental drivers making it extremely difficult to disentangle confounding influences. We have added the following caveat to the manuscript to convey that the relationship may not be causal: "However, it is possible that this correlation instead reflects correlated responses of emissions and stomatal aperture to increasing light and temperature."

*16L21-22. "A similar relationship between canopy-top methanol fluxes and concentrations is likely due to the influence of atmospheric concentrations on dry deposition to surfaces within the canopy." Similar comment. Temp/light would increase CH3OH fluxes which would in turn increase the concentrations. And wouldn't your point about deposition work in the opposite direction? At the lowest CH3OH concentration deposition should be lowest so that net emission is highest. But the data go in the opposite direction.

Thank you, that is very true. The model also includes a chemical sink, for which the same effect should be seen (i.e. the higher the concentration the higher the loss and the lower the gradient between in- and above-canopy concentrations, thereby reducing the flux). The positive correlation should therefore be ascribed to temperature/light effects increasing emissions at a greater rate than they increase loss processes. We have therefore re-worded this statement to read: "The positive relationship between canopy-top methanol fluxes and concentrations at low concentration is likely due to the

influence of increasing light and temperature increasing production of methanol at a greater rate than the loss processes (dry deposition to surfaces within the canopy and chemical loss). At higher concentrations, methanol loss rates increase sufficiently to balance production."

*"the level at which model and measurements are compared can also affect the measured-modeled bias, an effect compounded by the instantaneous nature of the model output fluxes". But aren't the model fluxes averaged over the same 30-min intervals as the data?

In this sentence, we were trying to describe how small changes in the vertical height can influence the comparison of measured-modeled fluxes. This statement has been revised to: "The heterogeneity of concentrations, concentration gradients and fluxes of methanol and acetaldehyde in time and space are evident from Fig. 5, demonstrating that the level at which model and measurements are compared can also affect the measured-modeled bias." This model artifact is in fact more of an issue for the more reactive bVOCs (e.g., isoprene), which have stronger concentration gradients. However, we feel it is still worth pointing out that there is a discrepancy in the height of the "canopy-top" between the observations and the model output.

*None of the simulations appear to capture the nighttime concentration decline for methanol. Is this a mixing effect or does it point to some shortcoming in the model treatment of deposition?

It does not appear to be a mixing effect as we constrain the mixing across the canopy top based on measurements. It is most likely due to the lack of an explicit treatment of wet deposition or wash-out within the model. We intend to develop FORCAsT in the future to include a representation of loss of methanol and other water-soluble compounds to wet surfaces based on relative humidity within the canopy as observations suggest this may be an important sink for such compounds.

*Perhaps I missed it, but if not please clarify how atmospheric mixed layer dynamics

are treated. Are those entirely prognostic within the model energy balance? How do we know how well the model captures the ML depth and growth/collapse timing, since those would clearly affect the diurnal concentration profiles that you interpret?

The following statement has been added to the manuscript: "Vertical mixing is calculated prognostically in the model following Blackadar (1979) and driven by top of canopy radiation and wind speed. The within-canopy wind profile is calculated following Baldocchi (1988). Turbulence and mixing in the canopy space is then modified according to Stroud et al. (2005) with wind speed and eddy diffusivity constrained to observations at the top of the canopy. A full description of the vertical mixing and its impact on concentration gradients is described in Bryan et al. (2012)."

We agree that BL mixing and particularly ML growth and decline are notoriously difficult to model and could account for the difficulties in reconciling observed fluxes and concentrations around dawn and dusk. Unfortunately this is also a period of time when flux measurements are subject to high levels of uncertainty making it hard to disentangle the various drivers. If anything the chaotic dynamics of the dawn/dusk transition periods strengthen our assessment that stomatal control is not significant at the canopy scale, as during these periods turbulence is the dominant driver of the observed exchange.

*Please double- check references.

Thank you for bringing that to our attention; we have checked and corrected the references.

Reviewer #2

We thank Reviewer #2 for their positive comments.

References

Baldocchi, D.: A multi-layer model for estimating sulfur dioxidedeposition to a deciduous oak forest canopy, Atmos. Environ.,22, 869–884, doi:10.1016/0004-

6981(88)90264-8, 1988.

Blackadar, A. K.: High-resolution models of the planetary boundary layer, in: Advances in Environmental Science and Engineering, 1, 50–85, Gordon and Breech Science Publishers, Inc., NewYork, USA, 1979.

Bryan, A. M., Bertman, S. B., Carroll, M. A., Dusanter, S., Edwards, G. D., Forkel, R., Griffith, S. Guenther, A. B., Hansen, R. F., Helmig, D., Jobson, B. T., Keutsch, F. N., Lefer, B. L., Pressley, S. N., Shepson, P. B., Stevens, P. S. and Steiner, A. L.: In-Canopy Gas-Phase Chemistry During CABINEX 2009: Sensitivity of a 1-D Canopy Model to Vertical Mixing and Isoprene Chemistry, Atmos. Chem. Phys., 12 (18), 8829-8849, doi:10.5194/acp-12-8829-2012, 2012.

Lee, B. H., Munger, J. W., Wofsy, S. C., and Goldstein, A. H.: Anthropogenic emissions of non-methane hydrocarbons in the north-eastern United States: Measured seasonal variations from 1992-1996 and 1999-2001, J. Geophys. Res., 111 (D20), D20307, doi: 10.1029/2005JD006172, 2006.

Parker, G. G.: Light Transmittance in a Northeastern Mixed Hardwood Canopy, Smithsonian Environmental Research Center Technical Report, 1998.

Stroud, C., Makar, P., Karl, T., Guenther, A., Geron, C., Turnipseed, A., Nemitz, E., Baker, B., Potosnak, M., and Fuentes, J. D.: Role of canopy-scale photochemistry in modifying biogenic atmosphere exchange of reactive terpene species: Results from the CELTIC field study, J. Geophys. Res., 110, D17303, doi:10.1029/2005JD005775, 2005.

Please also note the supplement to this comment:
http://www.atmos-chem-phys-discuss.net/acp-2016-522/acp-2016-522-AC1-supplement.zip

[Figure]

**Fig. 1.** Fig2_EnlargedScale

---

## Author Response (AR2)

Response to Co-Editor Decision:

I finally found the time to carefully check your response to the comments raised by the reviewers on your article on bi-directional exchange of methanol and acetaldehyde submitted for publication in

ACP. The comments by reviewer #1 tackle a number of issues and going through your response as well as the suggested revisions, it seems that this helps in overall arriving at an improved version of the ms. Consequently, I decided to accept the ms for publication in ACP after you have tackled this last round of minor editor comments just coming up reading your response and revisions.

In the revised ms version I noticed this feature that should be corrected: Pp 4, lines, 17-19, "While firmly based in plant physiology and plant response to environmental conditions, this approach would allow modelslackingleaf-level processestoaccountforthechanges in fluxdirection"

Thanks for picking that up. We have inserted spaces between the words.

Furthermore add the units after the term 3000, "where Rstom (($\mu$mol m-2 s-1)-1) is the stomatal resistance, 3000 (s m-1) is the model default limiting night time value of Rstom and n is a scaling factor"

Done

I have to say that this discussion on the selection of the values of the nocturnal stomatal resistance is not a strong feature of the paper also since it apparently is also included to represent removal over night by the (wet/dry) cuticle. Generally, choice of this nocturnal (stomatal) resistance term, which should reflect a general complete stomatal closure is rather arbitrary also due to the lack of direct observations. But having a quite high LAI can then even with such a selected large nocturnal value still result in quite some nocturnal removal and impact on simulated concentration also due to the limited mixing volume. Why did you not simply include the Rcut and Rstom terms as parallel resistances as is commonly done in dry deposition calculations and where Rstom (e.g., 1e6) is much higher than Rcut (3000 s m-1) resulting in a nocturnal Rleaf of 3000 s m-1 and a daytime Rleaf dominated by Rstom?

This is exactly how the model deals with "stomatal" resistance. It is the effective leaf resistance calculated from the parallel cuticle and stomatal resistances; at night the overall value is 3000 as the stomata are assumed fully closed. As the editor points out the night-time stomatal resistance is arbitrary and our initial choice of scaling factor was selected to be consistent with the resistance to dry deposition. Our sensitivity tests (in which we altered n) showed that this choice made no difference to our overall conclusion – that stomatal control of emissions at the leaf-level was not observable at the canopy-scale. We have re-phrased our explanation of the stomatal control factor in the text to read:

"The night-time "stomatal" resistance is in fact equal to the cuticular resistance and n was introduced to account for this. (During the day, the leaf resistance, the combination of the stomatal and cuticular resistances in parallel, is dominated by the stomatal resistance). The value of n was initially set to 3 for the S-storage and S-combo simulations, as Jarvis (1976) reported a limiting value of 1000 although this was species-dependent. The effect of the choice of value of $n$ is explored in Section 3.5."

In your response to the reviewers comment on the role of advection, I see your point about the potentially small impact of "anthropogenic" air masses with the winds generally coming from a direction with a small anthropogenic footprint. But what about having a footprint from a natural source region with also substantial biogenic sources of methanol and acetaldehyde? Would there be other ecosystems located more to the north that could also potentially have a large bVOC source?

While there could be large biogenic sources of methanol and acetaldehyde to the north of Harvard Forest, the inclusion of advected bVOCs did not improve the model-measurement fit. We conclude from this that there was little long-range transport from the north during the summer of 2012, i.e. that the winds were light and the site was impacted only by local sources of methanol and acetaldehyde with the same diel cycle as that emitted in-situ.

In your response to the reviewer on issues on the nocturnal decrease in methanol you mention that: "It is most likely due to the lack of an explicit treatment of wet deposition or wash-out within the model. We intend to develop FORCAsT in the future to include a representation of loss of methanol and other water-soluble compounds to wet surfaces based on relative humidity within the canopy as observations suggest this may be an important sink for such compounds". I agree with this point on the potential role of wet-canopy removal. This might be interpreted as a formality (also since you have not included a reference to this in the paper) but according to my long-term knowledge on the dry deposition process, removal of gases/aerosols by a wet canopy are also seen as a dry deposition process feature and not referred to a wet deposition or wash-out. This is really referring to the process of removal of gases and aerosols by scavenging and rain-out.

We acknowledge our choice of terminology was perhaps a little ambiguous; we were referring to removal of gases and aerosols by surface wetness within the canopy, a process that observations suggest may be important at some sites.

[revised manuscript text omitted]